# Large-scale forcing of the European Slope Current and associated inflows to the North Sea

Robert Marsh[1], Ivan D. Haigh[1], Stuart A. Cunningham[2], Mark E. Inall[2], Marie Porter[2], Ben I. Moat[3]

[1]Ocean and Earth Science, University of Southampton, National Oceanography Centre, Southampton, European Way, Southampton SO14 3ZH, UK
[2]Scottish Association for Marine Science, Scottish Marine Institute, Oban, Argyll, PA37 1QA, UK
[3]National Oceanography Centre, European Way, Southampton SO14 3ZH, UK

*Correspondence to*: R. Marsh (rm12@soton.ac.uk)

**Abstract.** The European Slope Current provides a shelf-edge conduit for Atlantic Water, a substantial fraction of which is destined for the northern North Sea, with implications for regional hydrography and ecosystems. Drifters drogued at 50 m in the European Slope Current at the Hebridean shelf break follow a wide range of pathways, indicating highly variable Atlantic inflow to the North Sea. Slope Current pathways, timescales and transports over 1988-2007 are further quantified in an eddy-resolving ocean model hindcast. Particle trajectories calculated with model currents indicate that Slope Current water is largely recruited from the eastern subpolar North Atlantic. Observations of absolute dynamic topography and climatological density support theoretical expectations that Slope Current transport is to first order associated with meridional density gradients in the eastern subpolar gyre, which support a geostrophic inflow towards the slope. In the model hindcast, Slope Current transport variability is dominated by abrupt 25-50% reductions of these density gradients over 1996-1998. Concurrent changes in wind forcing, expressed in terms of density gradients, act in the same sense to reduce Slope Current transport. This indicates that coordinated regional changes of buoyancy and wind forcing acted together to reduce Slope Current transport during the 1990s. Particle trajectories further show that 10-40% of Slope Current water is destined for the northern North Sea within 6 months of passing to the west of Scotland, with a general decline in this percentage over 1988-2007. Salinities in the Slope Current correspondingly decreased, evidenced in ocean analysis data. Further to the north, in the Atlantic Water conveyed by the Slope Current through the Faroe-Shetland Channel (FSC), salinity is observed to increase over this period while declining in the hindcast. The observed trend may have broadly compensated for a decline in the Atlantic inflow, limiting salinity changes in the northern North Sea during this period. Proxies for both Slope Current transport and Atlantic inflow to the North Sea are sought in sea level height differences across the FSC and between Shetland and the Scottish mainland (Wick). Variability of Slope Current transport on a wide range of timescales, from seasonal to multi-decadal, is implicit in sea level differences between Lerwick (Shetland) and Torshavn (Faroes), in both tide gauge records from 1957 and a longer model hindcast spanning 1958-2012. Wick-Lerwick sea level differences in tide gauge records from 1965 indicate considerable decadal variability in the Fair Isle Current transport that dominates Atlantic inflow to the northwest North Sea, while sea level differences in the hindcast are dominated by strong seasonal variability. Uncertainties in the Wick tide gauge record limits confidence in this proxy.

# 1 Introduction

The European Slope Current that lies to the west and north of Scotland exerts considerable influence on the physical and biogeochemical conditions on the adjacent west-European shelf seas (Huthnance et al., 2009), with Atlantic Water prevalent across much of the shelf (Inall et al., 2009). Located above the topographic slope at the eastern boundary, the Slope Current is associated with large-scale density gradients and wind forcing (Huthnance, 1984). Sea surface height drops in the northward direction, while prevailing wind stress is oriented from southwest to northeast. Density gradients and winds together drive eastward flows towards the slope that are diverted poleward as an intensified geostrophic flow along the slope. The barotropic transport of the Slope Current may be considered buoyancy-forced to first order, modified by frictional influences, with much of the seasonal variability in transport attributed to wind forcing (Huthnance, 1984).

The Slope Current is part of a greater inflow of Atlantic Water through the Faroe-Shetland Channel (FSC) (Sherwin et al., 2008; Richter et al., 2012; Berx et al., 2013) that also includes some recirculation of the Faroe Branch of Atlantic inflow, north of the Faroe Islands, that turns to flow southwestward, and an additional flow that has negotiated the Faroe Bank and Wyville Thomson Ridge. Sherwin et al. (2008) identify a long-term mean barotropic transport of 2.1 Sv over the upper part of the slope region of the Shetland shelf. Richter et al. (2012) refer to the flow between the Faroe and Shetland Islands as the Shetland Current, and use a range of tide gauge data to reconstruct transports in the region. However, they are unable to reconstruct Shetland Current transports, an issue that we return to in the discussion. Over 1995-2009, Berx et al. (2013) estimate an average net Atlantic inflow of 2.7 ± 0.5 Sv through the FSC. Calibrating sea level height with transport fluctuations, Berx et al. (2013) further use satellite altimetry to reconstruct volume transport since 1992, revealing a seasonal variation of 0.7 Sv in Atlantic inflow, becoming warmer and more saline since 1994, but with no trend in volume transport. However, this method detects the net inflow, not just that part associated with the Slope Current. As reviewed and discussed in Berx et al. (2013), issues remain with both the "altimetry transport" and the transport estimates based on ADCP and hydrography data, due to under-sampling of variability in time and space, and the extent to which either estimate is able to represent the net volume transport.

Beyond the FSC, most of the Atlantic inflow progresses beyond the Greenwich meridian to the Nordic seas, with a small fraction branching southward along the western flank of the Norwegian trench. Upstream of the FSC, Atlantic Water migrates up slope and onto the shelf through several processes, including wind forcing, frictional effects and flow instability related to topographic features (see Inall et al., 2009 and references therein). Major flow instability is associated with the Wyville Thomson Ridge, which presents a transverse obstacle to the Slope Current, bringing a substantial quantity of Atlantic Water onto the Shetland shelf (Souza et al., 2001), augmenting the on-shelf flows derived from further south (see Fig. 1 in Inall et al., 2009). The majority of this shelf flow turns into the North Sea between Orkney and Shetland as the Fair Isle Current (Dooley, 1974). The Fair Isle Current and southward flow along the flank of the Norwegian trench together comprise the Atlantic inflow to the North Sea, providing a relatively warm influence on the northern North Sea in winter. The North Sea as a whole has warmed considerably since the late 1980s, to an extent considered unprecedented in the

historical record (MacKenzie and Schiedek, 2007). Recent warming of the North Sea follows a wider pattern of warming across Europe, and increasingly mild winters in particular (MacKenzie and Schiedek 2007), although major inflows of warm Atlantic Water in 1988 and 1998 are also believed to have contributed to the warming (Reid et al., 2001), evident also in changes of zooplankton (Reid et al., 2003).

Episodic changes in Atlantic inflow have been attributed to anomalous wind forcing, hence wind-driven changes in Atlantic inflow and the associated warming have been a focus of recent model studies. Hjøllo et al. (2009) use a numerical model of the North Sea region to investigate changes of heat content over 1985-2007, in particular a long-term warming of 0.62°C. Dividing the North Sea into northern and southern circulation regimes, they find that inflows at the northern boundary are strongly influenced by large-scale atmospheric forcing associated with the North Atlantic Oscillation, but that variable

inflow at open boundaries has a limited direct influence on heat content variability. Winther and Johannessen (2006) relate changes of Atlantic inflow to wind forcing, but also emphasize the dilution of Atlantic Water as it circulates the North Sea before leaving in the Norwegian Coastal Current.

Interannual variability in the European Slope Current has recently been explored using altimetry data over a 20-year period, revealing a peak in poleward flow along much of the continental slope from Portugal to Scotland during 1995–1997, and a

long term decreasing trend of ~ 1% per year (Xu et al., 2015). Here, we consider the extent to which changes in the Atlantic inflow to the North Sea are associated with variability of the Slope Current driven by changing large-scale meridional density gradients and winds. We use a wide range of observations and eddy-resolving model hindcast data to examine the Slope Current, large-scale forcing mechanisms, and Atlantic inflow to the North Sea.

The paper is organized as follows. In Sect. 2, we outline the variety of data and methods used. In Sect. 3, we evaluate

simulated Slope Current drift over 1995-1997 using archived drifter data (Sect. 3.1). We then characterize Slope Current pathways, timescales and transports in a model hindcast spanning 1988-2007 (Sect. 3.2). With considerable Slope Current variability evident in the hindcast, we consider the influence of two large-scale driving mechanisms, meridional density gradients and wind forcing (Sect. 3.3), in both the model and observations. Finally, we explore the evidence for variable Atlantic inflow to the North Sea (Sect. 3.4), and sea level differences as proxies for Slope Current transport and this inflow

(Sect. 3.5). In discussion and conclusion (Sect. 4), we suggest that major variations in Slope Current transports and the Atlantic Water influence on the North Sea are primarily linked to variable meridional density gradients in the eastern subpolar gyre that are attributed to the combined (reinforcing) effects of wind and buoyancy forcing.

## 2. Datasets and Methodology

In Sect. 2.1, we summarize the available time series data that record variability in the Slope Current and the Atlantic inflow

to the North Sea over recent decades. In Sect. 2.2, we introduce the drifter data used to provide an observational perspective on the Slope Current system, and for preliminary evaluation of model currents. We then introduce the observations used to explore forcing mechanisms: mean absolute dynamic topography and climatological density data (Sect. 2.3); wind stress

reanalysis data (Sect. 2.4). In Sect. 2.5 we introduce the tide gauge data and analysis used to explore Slope Current transport between Shetland and the Faroes. Finally, we outline the model hindcasts used to characterize variability of the Slope Current system (Sect 2.6) and the Lagrangian diagnosis of hindcast data using the ARIANE methodology for calculation of particle trajectories based on velocity fields, and the accompanying statistical analyses (Sect. 2.7).

## 2.1 Time series data

The following data are provided as part of the ICES Report on Ocean Climate (IROC), available at http://ocean.ices.dk/iroc:

- Depth-averaged inflow and outflow to/from the North Sea, centred on 59°N, 1°E, as modelled volume transport between Orkney (Scotland) and Utsira (Norway), monthly averaged from January 1985
- Salinity in the Fair Isle Current, centred on 59°N, 2°W (first two stations on the JONSIS line), averaged over the depth range 0-100 m, irregularly-sampled and annually-averaged from 1960
- Salinity for the Faroe Shetland Channel – Shetland Shelf, centred on 61°N, 3°W, the maximum in the upper layer high salinity core, sampled 3 times per year (April/May, September/October and December) from 1950

We further sample monthly-mean salinity in the NCEP Global Ocean Data Assimilation System (GODAS) analysis fields spanning 1980-2016 (NOAA Climate Prediction Center, see http://www.cpc.ncep.noaa.gov/products/GODAS/). These time series data are used collectively to evaluate time series of similar quantities in the model hindcast and derived Lagrangian data.

## 2.2 Drifter data

As part of the Land-Ocean Interaction Study (LOIS), the Shelf Edge Study (SES) was undertaken in the mid-1990s. LOIS-SES included two Slope Current drifter experiments within which drifters were released in three groups of seven in an east-west line 20 km long across the continental shelf west of Scotland near 56.25°N, on 5 December 1995 and on 5-9 May 1996 (Burrows and Thorpe, 1999; Burrows et al., 1999), to characterize winter and summer conditions. The drifters were drogued at a depth of 50 m and tracked for up to 240 days, to study the regional circulation and dispersion. The archived drifter positions from both experiments are used to provide some context for the study, and for a basic evaluation of corresponding circulation in the model (see below).

## 2.3 Mapped Absolute Dynamic Topography (MADT) and climatological density

Daily global absolute sea-surface dynamic topography distributions with a spatial resolution of 0.25° are produced by Centre National d'Etudes Spatiales (CNES), and distributed through AVISO+ (http://www.aviso.altimetry.fr). Here, we use Delayed Time data from the SSALTO/DUACS system (AVISO+, 2014), which provides a homogeneous, inter-calibrated time series of sea-level anomaly. Absolute sea surface dynamic topography is the sum of sea level anomalies and a mean dynamic topography, both referenced over a twenty-year period (1993-2012). Key improvements in this new dataset are the use of a

new mean dynamic topography (MDT CNES_CLS13) calculated from GOCE satellite data, increased use of in-situ observations over the longer reference period, and more accurate mapping of the mesoscale (Rio et al., 2011). The geoid model developed from the GOCE satellite data has a horizontal resolution of 125 km. Multivariate objective analysis (including wind and in situ data) is used to improve the large-scale solution, resulting in a final gridded horizontal resolution of 0.25°.

Monthly estimates of ocean temperature and salinity spanning the same period are available as objectively-analysed gridded fields from the EN4 dataset provided by the UK Met Office Hadley Centre (Good et al., 2013). EN4 comprises global gridded fields of potential temperature and salinity at 1º resolution with 42 vertical levels. From 2002, the Argo float programme significantly improved EN4 data coverage in the northeast Atlantic (Good et al. 2013). The gridded temperature and salinity estimates are used to calculate climatological potential density referenced to the surface ($\sigma_0$), at selected depth levels. Of specific relevance to the Slope Current, and the present study, are meridional gradients of Mapped Absolute Dynamic Topography (MADT) and potential density.

**2.4 Wind stress data**

For the study period 1988-2007, we obtain 10-m winds from the ERA-interim 12-hourly, 0.75° x 0.75° resolution reanalysis datasets (Dee et al., 2011). We calculate wind stress following the methods of Large and Pond (1981). To address wind forcing of the Slope Current, a slope-based subset is extracted from the wind stress field between the 200 m and 1000 m contours (bathymetry from ETOPO1, Amante and Eakins, 2009) in the latitude range 48-60°N. These wind stress vectors are rotated into a coordinate system parallel to the 500 m contour and then averaged to obtain annual-mean averages over 1988-2007.

**2.5 Tide gauge data**

Monthly mean sea level records were obtained from the Permanent Service for Mean Sea Level (PSMSL; http://www.psmsl.org) for tide gauges at Wick (mainland Scotland; 3.09°W, 58.44°N), Lerwick (Shetland; 1.14°W, 60.15°N) and Torshavn (Faroes; 6.77°W, 62.01°N) (Holgate et al., 2013; PSMSL, 2015). The Wick record spans the period 1965-2014 and is 91.5% complete. The Lerwick record spans the period 1957-2014 and is 91% complete. The Torshavn record spans the period 1957-2006, and is 84% complete. Records have been corrected for the effects of glacial isostatic adjustment (GIA), using results from the ICE-5G model of post-glacial relative sea level history. We calculate Lerwick-Torshavn and Wick-Lerwick sea level differences as proxies for Slope Current transport and Fair Isle transport respectively. Note that the tide gauge records are referenced to different local datums and within the scope of this study it has not been possible to directly tie these together. However, as we are interested in transport variability, on seasonal to decadal timescales, we focus on the relative difference in sea level recorded by each tide gauge.

## 2.6 Model hindcasts

NEMO (Madec, 2008) is a state-of-the-art, portable ocean modelling framework developed by a consortium of European institutions. We sample currents and hydrographic data (temperature, salinity) from the northeast Atlantic region of an eddy-resolving (1/12°) global ocean model hindcast, the ORCA12 configuration of NEMO, for the period 1988-2007 (see Blaker et al., 2015), henceforth ORCA12-N01. With the barotropic Rossby radius at 55°N ranging from ~375 km (water depth 200 m) to ~1200 km (water depth 2000 m), the horizontal resolution of ORCA12 will comfortably resolve large instabilities and eddies associated with the Slope Current, although with corresponding baroclinic Rossby radii in the range 5-10 km, smaller-scale variability cannot be resolved. In the vertical dimension, there are 75 vertical levels, with 46 in the upper 1000 m, resolving the surface and bottom boundary layers that play an important role in Slope Current dynamics. The advantage of using fields from a global model is that large-scale influences on the Slope Current are fully represented, rather than being prescribed at the boundaries in a regional model, which can be problematic.

We use results for the hindcast period to simulate region-typical patterns of particle drift and dispersal (see methods). Our choice of this hindcast is guided by evidence that eddy-resolving simulations can faithfully reproduce the global EKE field observed with satellite altimetry (Petersen et al., 2013), while lower-resolution eddy-permitting simulations are known to substantially underestimate EKE (McClean et al., 2002; Hecht and Smith, 2013). NEMO is forced with 6-hourly winds supplied by the DFS4.1 (1988-2006) and DFS5.1.1 (2007-2010) datasets (Brodeau et al., 2010). The hindcast provides 5-day averages of currents and tracers (temperature, salinity), a time window appropriate to model realistically and with high precision the advection of an ensemble of particles representative of Slope Current transport, and associated variability on an eddy timescale of order 1 month. Most recently, a longer hindcast simulation with ORCA12 became available (e.g., Moat et al., 2016), henceforth ORCA12-N06, and we use diagnostics from this experiment to extend our analysis to the longer period 1958-2012. While it would be instructive to also calculate particle drift and dispersal with the longer hindcast, such calculations are not straightforward with the remotely archived ORCA12-N06 datasets.

## 2.7 Lagrangian model diagnostics

We use the ARIANE particle-tracking software (Blanke and Raynaud 1997) to track ensembles of particles that are "seeded" in the northward-flowing Slope Current. We release 630 particles, at 30 model levels from 9.85 m down to 371.22 m, and at 21 equally-spaced locations across a short section on the ORCA12 mesh (9.46°W 55.83°N to 9.28°W 55.82°N). This section is close to where floats have been deployed as part of the UK NERC project FASTNEt (http://www.sams.ac.uk/fastnet), henceforth the "FASTNET release section", and co-located with the location of the Slope Current in ORCA12, identified as a narrow band of high velocity (>10 cm s$^{-1}$) in 5-day mean fields. Particles are released on 1 January or 1 July, to sample the two halves of the seasonal cycle, with location, depth and ambient water properties (temperature, salinity) recorded every 24 hours for 183 days. We also use ARIANE in "backward" mode, which simply reverses (in time) the analytical calculation of particle progress through grid-cells, to examine the source of particles recruited to the Slope Current.

Particle locations are statistically analysed to obtain a measure of particle density, dividing the number of particle occurrences in a limited longitude-latitude range by the total number of particle occurrences during the tracking period. We use a 0.5° x 0.5° mesh to sample for particle occurrence, optimal for both the resolution of the Slope Current and sampling of sufficient particles from a statistical perspective. This quantifies interannual variation in pathways, broadly distinguishing

between years of low and high influence of the Slope Current on the northern North Sea, where we further record the presence of particles reaching the "NW North Sea" (south of 59°N, bounded by longitudes 4.5°W and 1.5°E) and the "NE North Sea" (east of 1.5°E, south of 62°N). Alongside particle density, we also obtain an average particle age (since release), depth and salinity, per 0.5° x 0.5° grid cell.

## 3. Results

We begin with a broad perspective of Slope Current pathways, moving on to examine time series of transport variability at selected locations. We then consider the drivers of transport variability, introducing a theoretical framework and applying this in an evaluation of the changes evident in our time series. We conclude the results section with a consideration of Atlantic inflow to the North Sea, and re-visit the prospects for monitoring regional transports with sea level observations.

### 3.1 Drifter observations and ORCA12 simulations of Slope Current pathways, 1995-97

To provide some context for the study, and a basic evaluation of corresponding model drift, in Fig. 1 we show LOIS-SES drifter data alongside example model particle trajectories, with the caveat that variability on length scales below ~10 km and time scales shorter than ~10 days are unresolved in the latter. LOIS-SES drifter deployments in December 1995 (Fig. 1a) and May 1996 (Fig. 1b) reveal somewhat different pathways in and around the Slope Current, with a tendency for more extensive drift in winter releases, compared to summer releases. Most drifters follow the Slope Current for several hundred

kilometres following release in December 1995. Several drifters enter the northeast sector of the North Sea by late winter or early spring, a travel timescale of 2-3 months. In contrast, several drifters released in May 1996 move onto the shelf and directly to the northwest North Sea, but on a wide range of timescales due to highly variable shelf currents.

Model particle trajectories start on 1 January and 1 July 1996 (Fig. 1c,d) at the FASTNEt release section (see Sect. 2.7), and are tracked forwards for 6 months. Particles released on 1 January tend to disperse more widely than those released on 1

July, with a larger number reaching the northwest and northeast sectors of the North Sea within 6 months. There are limitations to the direct comparison of the drifters and particle trajectories, as the former are subject to sub-mesoscale processes and tides that are not represented in the model. Given the chaotic nature of mesoscale variability, we further note that pathways inferred from a more limited number of drifters are less statistically significant. However, in broad terms, model particle trajectories indicate drift pathways and timescales similar to the drifters, and suggest more extensive drift in

the first half of 1996, compared to the second half, that is consistent with the observations and indicative of known seasonality in Slope Current dynamics.

## 3.2 Characterizing Slope Current pathways, timescales and transports in the ORCA12 hindcast of 1988-2007

As outlined in Sect. 2, particle density and mean age maps are obtained on a 0.5° x 0.5° mesh for each year from 1988-2007
(see Figs. S1, S2, S4 and S5 for 1 July starts, tracking forwards and backwards). The statistics for each set of 20 ensembles (January/July releases, tracked forwards/backwards) are further averaged to obtain the "grand ensemble" results shown in Figs. 2 and 3, respectively.

Tracking forwards, Fig. 2 shows the "grand mean" of particle density, age, depth and salinity, for particles released on 1 January (left panels) and 1 July (right panels). Ages are expressed as days since 1 January or 1 July. Particle density is
simply the fraction of all particle positions in each 0.5° x 0.5° grid square, in relation to all particle positions. With variation across three orders of magnitude, we use a log scale to highlight the distribution of density and depth statistics. Highest particle density (~0.1) and youngest age (0-20 days) is naturally located near the release section. Relatively high particle density in Fig. 2a,b otherwise traces the Slope Current pathway, characteristically following the shelf break (see Fig. 1), but bifurcating to the northeast of Scotland and just west of Norway. Only a small fraction of particles are tracked further to the
northwest and a destination in the Norwegian Sea. Back upstream, particles can also follow a minor pathway offshore to the north of the release section, turning westward to the south of Iceland and then southward along the Reykjanes Ridge. Another minor pathway involves almost immediate recirculation to the west of Ireland. Within 6 months, a few particles reach domain boundaries, to the north, east and west. With the focus of this study on the northern North Sea, these boundary terminations are not problematic.

Turning to the mean age of particles (Fig. 2c,d), this correspondingly increases to 140-180 days at locations most remote from the release section. There are relatively small differences between the January and July releases, although it appears that particles released in January reach the northwest North Sea more quickly, and in larger numbers. Mean ages for January releases are younger by ~10 days at many locations, suggestive of a more vigorous circulation during the first half of the year and more extensive shelf edge exchange, with higher on-shelf particle densities in particular.

Mean depths (Fig. 2e,f) are around 50 m in the North Sea inflow, which likely reflects the initial vertical distribution of particles in the Slope Current, but is consistent with residence of Atlantic Water in a sub-surface layer, below the fresh surface layer which is dominated by Baltic outflow. January releases reach slightly greater depths in the North Sea, compared to July releases, consistent with stronger southwesterly winds (aligned with the slope) and downwelling in winter/spring. Particles that leave the Slope Current system for an Atlantic fate are subducted across a wide range of depths,
up to around 1000 m, for both July and January releases. Particles that persist in the Atlantic inflow as far as the southern Nordic seas also descend on average, with mean depth of around 300 m.

Salinity in the Slope Current (Fig. 2g,h) is generally higher than surrounding waters. We consider the mean salinity of forward trajectories to trace high-salinity Atlantic Water through the Slope Current system. In the Faroe-Shetland Channel

(FSC), maximum salinity averages around 35.5. Water of this salinity corresponds to the "North Atlantic Water" of salinity around 35.42 psu that dominates the upper 200 m on the Shetland side of the FSC, recently identified by McKenna et al. (2016). Moving onto the Shetland shelf, upstream of FSC, mean salinity declines to around 35.3 psu. This is consistent with mixing of Atlantic Water and relatively fresher water on the shelf, for which salinity ranges 35.00-35.25 psu at the Ellett line
(e.g. Fig. 14 in Inall et al. 2009).

Tracking backwards, Fig. 3 shows the corresponding particle density, age and depth distributions for flows feeding transport across the FASTNEt release section. Similar to the results for forward tracking, highest particle densities are located adjacent to this section. In contrast to the forward tracking, particle density is generally lower across a broader area of the eastern subpolar gyre, indicative of a widespread inflow across the approximate latitude range 48-60°N. As for forward tracking, a
few back trajectories reach the western domain boundaries in a little under 180 days. There is some evidence for a southward continuation of the Slope Current, along the shelf break, to around 13°W, 48°N. This is more evident in back-trajectories that span the second half of the year (i.e., reaching the FASTNEt release section in January). This "upstream" branch of the Slope Current is slower than the "downstream" branch represented in Fig. 2, consistent with downstream strengthening of Slope Current transport through progressive inflow from the west. Mean depth across the catchment area generally increases to the
south, with an impression that the upper ~100 m layer of the Slope Current is recruited from the southeast subpolar gyre, while deeper layers (below 100 m) are recruited from the northeast subtropical gyre. Particles arriving in the Slope Current in January originate from a depth range 10-50 m in the subpolar gyre, while particles arriving in July arrive from greater depths in this region (around 50-100 m), indicating stronger upwelling of particles recruiting to the Slope Current during the first half of the year.

We now consider the corresponding changes in Slope Current transport at selected locations along the shelf break. Figure 4 shows Slope Current transport every 5 days over 1988-2007 at a somewhat longer FASTNEt section (from 9.74°W, 55.82°N to 9.28°W, 55.79°N), at two further sections – EEL and Shetland Slope – and for Atlantic inflow to the North Sea in the Fair Isle Current, (see Fig. 1c,d). End-points for the EEL section (from 9.48°W, 57.11°N to 8.57°W, 57.05°N) and the Shetland Slope section (from 2.72°W, 60.87°N to 2.15°W, 60.57°N) are based on particle trajectories (see Fig. 1), rather than strictly
delimited by the same isobaths (model bathymetry) spanned at the FASTNEt section (181 m to 1516 m). Hence the EEL section spans a depth range 120-1046 m, while the Shetland Slope section spans 133-412 m. On this basis, we diagnose Slope Currents transports at the upper end of observed ranges (e.g., Sherwin et al., 2008), and considerably higher than "Atlantic inflow" estimates of $2.7 \pm 0.5$ Sv in the Faroe-Shetland Channel (Berx et al., 2013). Between the FASTNEt (EEL) and Shetland Slope sections, long-term mean transport increases, while the standard deviation decreases, from $4.62 \pm 3.34$
$(4.17 \pm 2.71)$ Sv to $7.04 \pm 2.18$ Sv (Table 1). The Fair Isle Current, an inshore component of Slope Current transport, amounts to $1.17 \pm 1.11$ Sv. The relatively large standard deviation results from strong seasonality, with peak inflow in winter.

As a metric of seasonal variations in transport, we sample the ensemble-mean particle ages in Fig. 2 for travel times between sections. These are shorter and less variable in the first half of the year: $19.4 \pm 7.7$ days (January releases) compared to 28.7

± 20.4 days (July releases) between FASTNEt and EEL sections; 90.4 ± 23.7 days (January releases) compared to 98.6 ± 32.3 days (July releases) between FASTNEt and Shetland Slope sections; 123.6 ± 20.0 days (January releases) compared to 145.6 ± 14.6 days (July releases) between FASTNEt and the Fair Isle Current sections (Table 1). This is indicative of a somewhat more vigorous circulation during January-June, although monthly-mean transports at the three Slope Current sections and the JONSIS section (Fig. S6) do not provide conclusive evidence for this.

Regarding the 1988-2007 variability in Fig. 4, transports are weaker by 9-45% in the second decade of the hindcast at all sections (see Table 1), with a most striking shift to weaker transport at the EEL section over 1996-1998. To investigate the extent to which Slope Current transport variability (including the seasonal cycle) is instantaneously correlated at the three sections, we compute correlation coefficients between the 5-day averaged transports in Fig. 4, confirming that variability along the shelf break is coordinated to a large extent. This is most evident between the FASTNEt and EEL sections, for which the correlation is 0.64 (significant at 99% confidence level), as might be expected given the relatively short distance separating these two sections. More striking is a correlation of 0.43 (significant at 99% confidence level) between transports at the widely separated EEL and Shetland Slope sections.

## 3.3 Mechanisms driving Slope Current variability

We now consider in turn the influences of meridional density gradients to the west of the shelf break, and the local winds along the shelf break, in driving the variability in Slope Current transport evident in the ORCA12-N01 hindcast.

### 3.3.1 Meridional density gradients

To first order, the Slope Current is driven by the deep ocean meridional density gradient. We can accommodate this in the geostrophic momentum balance, as presented in Simpson and Sharples (2012) and reproduced here. First consider the zonal momentum equation, given reference density $\rho_0$ and Coriolis parameter $f$. We use the hydrostatic balance, whereby pressure $p = \rho g(z + \eta)$, given density $\rho$, gravitational acceleration $g$, arbitrary ocean depth $z$ and sea surface elevation $\eta$, and we assume that the zonal density gradient is zero. The right hand side thus simplifies to a zonal gradient in sea surface height:

$$-fv = -\frac{1}{\rho_0}\frac{\partial p}{\partial x} = -\frac{1}{\rho_0}\frac{\partial(\rho g(z+\eta))}{\partial x} = -g\frac{\partial \eta}{\partial x}, \tag{1}$$

Vertically integrating in the depth range $-h$ and $\eta$, defining meridional transport, $V = \int_{-h}^{\eta} v dz$, and assuming $h \gg \eta$, the depth-integrated zonal momentum balance follows as:

$$-fV = -g\frac{\partial \eta}{\partial x}h, \tag{2}$$

Considering the y-momentum equation, we follow the same approach, noting that the meridional density gradient is non-zero, so the right hand side now includes an extra term:

$$fu = -\frac{1}{\rho_0}\frac{\partial p}{\partial y} = -\frac{1}{\rho_0}\frac{\partial(\rho g(z+\eta))}{\partial y} = -\frac{g}{\rho_0}\frac{\partial\rho}{\partial y}z - g\frac{\partial\eta}{\partial y}, \tag{3}$$

Vertically integrating again, defining zonal transport, $U = \int_{-h}^{\eta} u\,dz$, the meridional momentum balance follows as:

$$fU = -\frac{g}{2\rho_0}\frac{\partial\rho}{\partial y}h^2 - g\frac{\partial\eta}{\partial y}h, \tag{4}$$

Cross-differentiating (2) and (4) for $\partial U/\partial x$ and $\partial V/\partial y$, and given vertically-integrated continuity of volume, $\partial U/\partial x +$
$\partial V/\partial y = 0$, we obtain an expression for the meridional gradient in sea surface elevation as a function of local depth ($h$) and the meridional density gradient:

$$\frac{\partial\eta}{\partial y} = -\frac{h}{\rho}\frac{\partial\rho}{\partial y}, \tag{5}$$

Following Simpson and Sharples (2012), we further distinguish between the shelf (depth $h = h_s$) and the deep ocean ($h = H$):

Shelf: $\qquad\qquad \left(\frac{\partial\eta}{\partial y}\right)_{h_s} = -\frac{h_s}{\rho}\frac{\partial\rho}{\partial y}, \tag{6}$

Deep ocean: $\qquad \left(\frac{\partial\eta}{\partial y}\right)_{H} = -\frac{H}{\rho}\frac{\partial\rho}{\partial y}, \tag{7}$

Applying this theoretical framework, we use MADT and climatological temperature and salinity observations to evaluate the meridional gradients of Eq. (7) in the eastern subpolar North Atlantic, where the Slope Current originates (see Fig. 3). Fields of MADT and mean $\sigma_0$ at 500m (Fig. S7a,b) are broadly characterized by negative and positive meridional gradients respectively (Fig. S7c,d). Dividing $\partial\eta/\partial y$ by $-\rho^{-1}\,\partial\rho/\partial y$, we obtain an estimate of the deep ocean depth scale $H$, plotted in Fig. 5. Over large areas of the region, $H$ is thus predicted in the range 500-2500 m, representative of the deep ocean.

Since $H \gg h_s$, Eqs. (6) and (7) predict that the (downward) meridional gradient in $\eta$ will be greater over the deep ocean than over the shelf, so the cross-slope (downward) gradient in $\eta$ increases with latitude (see also equation pair 10.11 in Simpson and Sharples, 2012). This cross-slope difference in sea surface elevation will result in a geostrophic current parallel to the isobaths. At the same time, the momentum balance in the meridional direction implies a geostrophic transport in deep water towards the slope, predicted by substituting Eq. (7) into Eq. (4):

$$U = \frac{gH^2}{2\rho_0 f}\frac{\partial\rho}{\partial y}, \tag{8}$$

As this zonal flow reaches the slope, it turns to the north and joins the meridional current. This current increases with latitude as the zonal difference in height across the sloping seabed, which increases likewise (see also equation 10.12 in Simpson and Sharples, 2012). Using Eq. (8) with g = 9.81 m$^2$ s$^{-1}$, $\rho_0$ = 1025 kg m$^{-3}$, representative latitude 55°N (f ~ 1.19 x 10$^{-4}$ s$^{-1}$), and $H$ = 1000 m as a depth scale appropriate for the inflow (from Fig. 5), we obtain $U \sim 8$ x 10$^7$ $\partial\rho/\partial y$, (m$^3$ s$^{-1}$ per m along slope). Considering the Slope Current to be thus "fed from the west" by geostrophic inflows that are supported by a meridional density gradient, we now investigate how this large-scale pattern may have changed over the hindcast period. Supplementary

Figs. S8 and S9 show maps of potential density ($\sigma_0$) for 1-5 January, biennially over 1988-2006, at two depth levels - 500 m and 947 m - which are representative of the inflow. At 500 m, it is evident that density to the south of Iceland progressively decreases over 1988-2006; at 947 m there is a progressive increase of density at mid-latitudes over the study period, with little change further to the north. Both changes result in a reduction of the northward density gradient.

Variability of $\partial\rho/\partial y$ is more explicitly shown in Fig. 6 as northward trends of $\sigma_0$ over the latitude range 45-62°N, in the northeast Atlantic (across 15-28°W), 5-daily for 1988-2007, at 500 m (Fig. 6a) and at 947 m (Fig. 6b). Reductions in the trend are evident throughout most of the period, with particularly abrupt reductions over 1995-1997 and overall reductions in the range 25-50%, with strongest % reductions in the east (around 15°W). These abrupt changes are consistent with the sharp reduction of Slope Current transport at the FASTNEt and EEL sections (Fig. 4a,b).

As a metric for density forcing of the Slope Current, we average the 45-62°N density gradients across 15-28°W and annually, and then take annual anomalies relative to the 20-year mean. Figure 7 shows 1988-2007 time series of these metrics for density gradients at 500 m (Fig. 7a) and at 947 m (Fig. 7b). The annual index clearly shows how the density gradients weakened around the mid-1990s. Taking a change in $\partial\rho/\partial y$ of 2-4 x $10^{-3}$ kg m$^{-3}$ degree$^{-1}$ (111 km), Eq. (8) suggests a change of inflow, $\Delta U \sim 8$ x $10^7$ x [2-4] x $10^{-3}$ x (111 x $10^3$)$^{-1}$ = 1.42-2.9 m$^2$ s$^{-1}$. Across 17° of latitude, this amounts
to a change (decrease) of total inflow in the range 2.75-5.5 Sv, broadly consistent with the abrupt drop in transport, over 1996-1998, at the EEL section in particular (Fig. 4b).

In Fig. 7c and 7d, we plot these metrics against annual-mean SC transports at the EEL section. We find strong and significant correlations, of up to 0.67 and 0.75 (both significant at 99% confidence level) between Slope Current transport at the EEL section and the density gradient indices, at 509 m and 947 m respectively. The strong correlations are associated
with a degree of bimodal scatter in Figs. 7c,d, associated in turn with abrupt declines of density gradients (Fig. 7a,b) and Slope Current transport (Fig. 4b) in the mid 1990s. Similar correlations, also significant at 99% confidence level, are obtained between the density gradient indices and annual-mean transports at the FASTNEt section (0.67 at 509 m; 0.71 at 947 m) and at the Shetland Slope section (0.61 at 509 m; 0.60 at 947 m).

### 3.3.2 Wind forcing

While density gradients do indeed appear to exert a leading control on barotropic Slope Current transport, wind forcing is also likely to play an important role on short timescales. In particular, strong wintertime winds likely explain strongest Slope Current transport at that time of the year (Huthnance, 1984). For a circular basin with a sloping margin and wind-stress forcing, Huthnance (1984) uses scaling arguments applied to incompressible, hydrostatic momentum equations (with horizontal flow scales > topographic slope scale and vertical scales; and w << u) to demonstrate that in a steady state, any
component of wind stress parallel to a steep slope ($\tau^s$) will induce a downwind current along the continental shelf and slope, with a speed given by $\tau^s/(\bar{\rho}k)$, where $\bar{\rho}$ is a depth mean density and $k$ is a linearized friction coefficient. Additionally,

applying a uniform azimuthal density gradient, and similar scaling arguments, a slope/shelf current (in the direction of decreasing sea surface height) will have strength comparable to the wind-stress induced current if:

$$\frac{\partial \rho}{\partial s} = \frac{A}{hHg}|\tau^s| ,$$ (9)

where $\partial\rho/\partial s$ expresses the depth-mean along-slope density gradient, $A = 2$ to $4$ (dependent on scaling assumptions) and $h$
and $H$ are the local and maximum water depths (see Huthnance 1984, un-numbered equation, end of p.799).

In interpreting these scaling arguments as applied to the eastern margin of the North Atlantic, the underlying physics is such that in a steady sense both the eastward geostrophic flow (derived from the meridional density gradient), and an eastward surface Ekman response (derived from a northward wind-stress) drive water initially towards the closed eastern boundary. These eastward flows raise the sea level near the eastern boundary, and result in a geostrophically balanced northward flow,
with friction balancing the down-wind and down-pressure gradient accelerations.

Although unimportant when making a relative comparison between wind stress and buoyancy forcing, the linearized friction coefficient determines the absolute strength of the northward current (for given forcing). Expressed as $k \propto U_{M2}/h$ (where $U_{M2}$ is a magnitude for the semi-diurnal tidal current speed), friction therefore determines the zonal structure of a Slope Current (moving across the slope). In the setting of the shallow and strongly tidal northwest European shelf, friction is
greatest on the shelf, and the strongest northward flows are therefore concentrated over the slope.

The effects of seasonality in a northeast Atlantic setting are noteworthy. In winter, the surface Ekman layer will be deeper than the shelf break, and the eastward Ekman mass convergence will manifest at least in part over the continental slope. Both eastward Ekman and eastward geostrophic flow (in balance with the meridional density gradient) therefore have co-located convergence over the slope in winter, forcing a strong and mostly barotropic Slope Current. In summer, by contrast, the
surface Ekman layer will be considerably shallower than the shelf break, with convergence occurring more towards the coast. This leads to greater surface water exchange onto the shelf, and a more spatially diffuse northward flow over the slope and shelf, since the effects of wind-stress and meridional buoyancy forcing are no longer co-located over the slope.

Here, we focus on interannual changes in wind forcing, considering the influence on Slope Current transport of anomalies in along-slope wind stress. As a metric for the wind-stress forcing of the Slope Current, and for direct comparison with the
density gradient metric developed in Sect. 3.3.1, we evaluate the right hand side of Eq. (9) annually over the continental slope between 48-62°N, where $|\tau_s|$ is obtained from re-analysis 10 m winds, as outlined in Sect. 2.4. Expressed in units of northward density gradient ($10^{-3}$ kg m$^{-3}$ degree$^{-1}$, regressed in the latitude range 48-62°N), annual anomalies of this metric are shown in Fig. 8. The anomalies are generally smaller in magnitude compared to the large-scale anomalies in Fig. 7, but there is a similar tendency for positive (negative) anomalies before (after) 1996 (as Fig. 7). In Sect. 4, we discuss the
combined influences of wind and buoyancy forcing on variable Slope Current transport, as diagnosed with this common framework.

**3.4 Atlantic inflow to the North Sea**

Considering the Slope Current and associated flows through the Faroe-Shetland Channel (FSC), we identify that part of the flow diverted as Atlantic inflow to the North Sea. Time series of the associated transports in ORCA12-N01 are presented in Figure 9. Net transports between Faroes and the Scottish mainland at Wick (not shown) are very highly correlated with net transport between Faroes and Shetland (r = 0.98). Differences between transport across Faroes-Wick and Faroes-Shetland sections are due to net flow between Shetland and Wick (Fig. 9a). The sign convention of this residual transport as plotted is positive into the North Sea. This residual flow alternates between typically positive values in winter and negative values in summer. Across a section from 1.10°W to 2.46°W at 59.27°N (the western JONSIS line), we obtain transports representative of the Fair Isle Current (Fig. 9b). This transport is almost always positive, i.e., into the North Sea, peaking in winter with seasonal amplitude very similar to that seen in Fig. 9a. Given the varying sign of residual flow and persistent southward transport in the Fair Isle Current, we infer a steady recirculation around Shetland of ~1 Sv.

To emphasize the dominant contribution of Fair Isle Current fluctuations to the variability of Atlantic inflow, we co-plot 30-day running means of anomalies in inferred Atlantic inflow and Fair Isle Current transport (Fig. 9c). These time series are highly correlated: r = 0.81; significant at 99% confidence level. To evaluate the realism of this part of the Atlantic inflow in ORCA12, Fig. 10a shows monthly Atlantic inflow estimates from the ICES Report on Ocean Climate alongside a 30-day running mean of the 5-day transports in Fig. 9b. Close correspondence between the two time series is again consistent with dominance of Atlantic inflow by the Fair Isle Current. In Fig. 10b and 10c, we show annual-mean transport anomalies from the ICES Atlantic inflow and ORCA12 Fair Isle Current transport, both relative to the 1988-2007 mean. It appears that transports declined by 0.3-0.4 Sv over the 1990s, corresponding to around 20% of the mean flow.

Returning to the Lagrangian analysis, and noting the tendency for particles to separately branch into the northwest and northeast North Sea (see Fig. 2), we consider the percentage of particle counts in "NW North Sea" and "NE North Sea" sub-regions (see Sect. 2.7 for definitions). For each year over 1988-2007, we average these statistics for both January and July releases. Figure 11 shows histograms of this annual mean %, for each sub-region (Figs. 11a,b) and both combined (Fig. 11c). In terms of a combined presence in the North Sea, the % of particles released at the FASTNEt section declines from near 40% in the early 1990s to around 15% in the mid 2000s. This long-term decline is suggestive of a reduced influence of Slope Current water of Atlantic origin in the northern North Sea, in the ORCA12 hindcast. The decline is similarly evident in the separate January and July releases (not shown), indicating a year-round character.

Finally, we consider salinity as a tracer of Atlantic Water, evaluating salinity variations in the hindcast alongside available observations. Sampling salinity along each model trajectory, we obtain averages per 0.5° x 0.5° grid cell (see Fig. S3) specific to time-varying flows at our four selected sections. We sample the NCEP Global Ocean Data Assimilation System (GODAS) analysis fields at locations central to each section (see Sect. 2.1), to obtain monthly mean salinity in that part of the water column most influenced by Atlantic Water (see Supplementary Fig. S10), subtracting climatological seasonal cycles to obtain time series of salinity anomalies (see Supplementary Fig. S11). We also consider direct observations, synthesized by ICES, where the Slope Current and Atlantic inflow have been monitored since 1950 and 1960 respectively:

annual-mean observed salinity of Atlantic Water for the Faroe Shetland Channel – Shetland Shelf; annually and vertically averaged salinity in the Fair Isle Current (see Sect. 2.1). These salinity data are presented in Figure 12.

At the FASTNEt and EEL sections (Figs. 12a,b), there is an overall decline of salinity from the early 1990s to the mid 2000s, in both ORCA12 and GODAS data. Superimposed on these declines is notable interannual variability, more dominant in the GODAS data, which also indicate a reversion to increasing trends from the mid-2000s onwards. At the Shetland Slope, long-term observed increases of salinity from 1980 to the early 2000s are evident in both GODAS and ICES data, while a declining trend persists in ORCA12 (Fig. 12c). We suspect that increasing trends observed in the Faroe Shetland Channel are associated with an additional influence from oceanic Atlantic Water (separate from Slope Current traversing FASTNEt and EEL sections) that is not well represented in the hindcast. At the western JONSIS line, a slight freshening trend is evident from the mid 1990s to the mid 2000s, in the direct observations, the GODAS analysis and ORCA12, although interannual variability is considerable in the observations but much reduced in ORCA12. The observed variability may be associated with local processes that are under-represented in ORCA12. We note a remarkable increase of salinity around 2008 in the GODAS data (see also Fig. S11d), positive anomalies being sustained up to 2016.

## 3.5 Sea level differences as proxies for Slope Current transport and Atlantic inflow to the North Sea

As Slope Current transport is strongly barotropic, we expect a strong correlation with the sea surface difference across the current. Given the available tide gauge data at Wick, Lerwick and Torshavn (Sect. 2.5), we consider the differences in relative sea surface height (SSH) between Shetland and the Faroes (Lerwick-Torshavn), and between mainland Scotland and Shetland (Wick-Lerwick), in ORCA12, presented in Figure 13. For the 1988-2007 hindcast, Fig. 13a shows 5-day averages of SSH at the nearest ocean grid cells to Wick (green curve), Lerwick (red curve) and Torshavn (blue curve), and the differences, Lerwick minus Torshavn (Fig. 13b) and Wick minus Lerwick (Fig. 13c). There is a clear seasonal cycle in SSH at all three locations, with higher (lower) SSH in summer (winter), largely due to the thermosteric effect of winter cooling (summer warming) and contraction (expansion) of water columns. The seasonal cycle increases in amplitude from Torshavn to Wick, hence there is also a seasonal cycle in the SSH difference, with larger differences in winter. Comparing Fig. 13b and Fig. 4c, seasonal cycles of Shetland Slope transport and Lerwick-Torshavn SSH difference are clearly in phase, as are Fair Isle Current transports (Fig. 9b) and Wick-Lerwick SSH differences (Fig. 13c).

Removing mean seasonal cycles in transports and SSH differences from 5-day averaged data, we find that transport anomalies are strongly correlated with SSH difference anomalies: $r = 0.68$ for Shetland Slope transports and Lerwick-Torshavn SSH differences; $r = 0.85$ for Fair Isle Current transports and Wick-Lerwick SSH differences; both significant at 99% confidence level. Linear regressions indicate a transport sensitivity of ~0.25 Sv per cm. Illustrating this sensitivity, Figure 14 shows 30-day running averages of anomalies, relative to seasonal cycles over 1988-2007, for Shetland Slope transport and Lerwick-Torshavn SSH difference (Fig. 14a,b), and for Fair Isle Current transport and Wick-Lerwick SSH difference (Fig. 14c,d). SSH differences may therefore be useful proxies for Slope Current transport and Atlantic inflow to the North Sea.

To relate changes in Atlantic inflow to the North Sea with changes in Slope Current transport and SSH difference, we correlate the "combined" % of North Sea particles (Fig. 11c) with annual-mean anomalies for Shetland Slope transport and Lerwick-Torshavn sea level difference. Correlation coefficients of 0.52 (with transports) and 0.70 (with SSH differences), both significant at 99% confidence level, indicate that larger SSH differences and stronger transports are indeed associated

with more Atlantic Water reaching the North Sea. The stronger correlation of % North Sea particles with SSH difference (compared to transport) indicates that this metric more completely captures transport variability than the short and fixed Shetland Shelf section. Clearly then, declining Slope Current transport at the Shetland Slope over 1988-2007 is broadly representative of changes already evident in the wider (upstream) Slope Current system, and consistent with the declining % of Slope Current water particles reaching the North Sea.

Evidence for variable Slope Current transport over a longer time period is now considered, using historical tide gauge data and the longer ORCA12-N06 hindcast. Figure 15 shows sea level at Wick, Lerwick and Torshavn from 1957 onwards in tide gauge records, monthly averaged after correcting for GIA (Fig. 15a), 30-day running means of SSH over 1958-2013 in the ORCA12-N06 hindcast (Fig. 15b), and the corresponding differences (Fig. 15c,d). Considering the sea level time series (Fig. 15a,b), there is general agreement between the tide gauges and ORCA12 in terms of a seasonal cycle and long-term sea level

rise primarily associated with thermal expansion, although irreconcilable differences currently remain between the datum levels for the two tide gauge records (therefore Fig. 15a and Fig. 15b should not be directly compared).

Considering Lerwick-Torshavn sea level differences in the model (Fig. 15c,d), ORCA12-N06 indicates a degree of low-frequency variability, with smaller differences in the last ~15 years that are in close agreement with ORCA12-N01 over the period of overlap (see Supplementary Material, Fig. S12): averaged over 1989-99 and 1999-2012, sea level differences in the

hindcast are 15.0 cm and 13.1 cm respectively, equating to a transport reduction of around 0.5 Sv in the latter period. This reduction in the sea level differences is seen to an extent in the tide gauge records whenever the data is available. However, substantially higher differences over 1957-64 apparent from the tide gauge records are not seen in the hindcast. Considering Wick-Lerwick sea level differences, higher amplitude variability on decadal timescales is apparent in the tide gauge data, compared to the hindcast. Considering the period 1965 onwards, when Wick tide gauge data are available, correlation

coefficients between monthly tide gauge sea level differences and model SSH differences are strong for Lerwick-Torshavn ($r = 0.61$, significant at 99% confidence level), but weak for Wick-Lerwick ($r = 0.11$, significant at 95% confidence level).

As related to variability in sea level differences, Atlantic inflow to the North Sea is strongly correlated with FSC Slope Current transport in the hindcast, ($r = 0.71$ for Lerwick-Torshavn and Wick-Lerwick SSH differences, significant at 99% confidence level). The corresponding tide gauge differences are, however, not correlated ($r = -0.05$, not significant),

indicating that other factors influence the majority of observed sea level variability at Wick in particular. These preliminary findings help to validate the Slope Current variability simulated in ORCA12 hindcasts, while indicating the limited extent to which variable Slope Current transport and Atlantic inflow to the North Sea may be reconstructed with tide gauge data over the longer historical era. To obtain a useful proxy for Atlantic inflow, it will be necessary to first remove that part of the variability in the Wick tide gauge record that not associated with dynamical signal.

## 4. Discussion and Conclusions

The Slope Current system that is observed to follow the shelf break to the west and north of Scotland is investigated using a range of observations and an eddy-resolving ocean model (ORCA12) hindcast spanning 1988-2007. Deployments of drogued drifters over 1995-1997 reveal a variety of pathways and timescales in the Slope Current system, hinting at seasonal to interannual variations. To further explore this variability, offline particle trajectories are calculated with model currents. Particles are tracked both forwards and backwards in time, for 183 days, from a section across the Slope Current (9.46-9.28°W at ~55.82°N, in the upper 371 m) where floats have been deployed as part of the UK NERC project FASTNEt (http://www.sams.ac.uk/fastnet). Tracked backwards, particle trajectories reveal a major source of Slope Current water in the eastern subpolar gyre, with a smaller proportion advecting with the Slope Current from more southern latitudes.

Variable pathways are related to both seasonal and interannual variability in Slope Current transport. The latter variability is related to large-scale forcing mechanisms. Downward trends in Slope Current transport, similar to those inferred from altimetry (Xu et al., 2015), are principally related to basin-scale changes in the subpolar North Atlantic. Across the northeast Atlantic over 1988-2007, we identify 25-50% reductions of meridional density gradients in the depth range 500-1000 m representative of a layer that supports geostrophic inflow to the Slope Current, which can be considered as "fed from the west". In particular, we find abrupt reductions of density gradients over 1995-97, coincident with weakening of the Slope Current at the FASTNEt and EEL sections. The reductions in meridional density gradients are primarily due to warming in the eastern subpolar gyre, coincident with weakening of the subpolar gyre (Johnson et al., 2013). The new OSNAP monitoring array (http://www.o-snap.org/), spanning the subpolar gyre and incorporating the EEL section, should provide the observations needed to further investigate large-scale drivers of variability in Slope Current transport.

Using a common framework, we find that changes in annual-mean wind forcing contribute around 20% to the density gradient variability. Changes in wind forcing, specifically the along-slope component of wind stress, are associated with the transition of the North Atlantic Oscillation (NAO) from a positive to a neutral phase during the mid-1990s, weakening wind forcing of the Slope Current at that time. To summarize forcing mechanisms, the schematics in Fig. 16 indicate the density gradients, wind forcing, Ekman transports and sea surface slopes associated with weak and strong Slope Current transport. Strong (weak) transport is associated with a strong (weak) subpolar gyre, and the NAO in a positive (negative) phase. Emphasizing the conditions for strong Slope Current transport: in the deep ocean, colder water to the north sets up a stronger northward density gradient, while the downward sea surface slope to the north steepens; stronger eastward geostrophic flow is supported by the combined effect of density gradient and strengthened along-slope (northward) winds and onshore Ekman transports; northward steepening of the cross-slope gradient in sea surface height becomes more pronounced in proportion to inflow recruited to the barotropic Slope Current.

Downstream consequences of changes in Slope Current transport have also been investigated. Tracked forwards in the ORCA12 hindcast, a substantial number of particle trajectories reach the northern North Sea, and we accordingly diagnose the % of particle locations in the northwest and northeast North Sea, as metrics for cumulative Atlantic inflow in these

regions. Over the 1988-2007 hindcast, we thus identify a decline from ~40% in the early 1990s to ~15% in the mid-2000s, accompanied by the reductions in Slope Current transport. Around half of the Atlantic inflow is mixed with fresher North Sea water (including Baltic outflow) before outflow in the Norwegian Coastal Current (Winther and Johannessen, 2006). Mean salinities along particle trajectories indicate a reduction of 0.2-0.3 psu from inflow to outflow (see Fig. S3). Variable

Atlantic and Baltic inflows must contribute to salinity variability in the North Sea. There is a climatological seasonal variation of North Sea freshwater content by ~20%, with peak values in July/August that lag by 2-3 months the net fresh inflow from the Baltic, which in turn varies by a factor of ~3 over the seasonal cycle (see Sündermann and Pohlmann, 2011 - their Fig. 17). While Baltic inflow dominates the seasonal cycle of salinity, Atlantic inflow is thought to dominate mean salinity of the North Sea (Sündermann and Pohlmann, 2011).

Changes in salinity of the Atlantic inflow, on interannual and longer timescales, are also likely to impact North Sea salinity. While observed increases of Atlantic Water salinity in the Faroe-Shetland Channel over 1988-2007 (Holliday et al., 2008) are not reproduced in the hindcast, observed salinity in the Fair Isle Current is highly variable, and salinity remained relatively invariant in the northern North Sea (Larsen et al., 2016). At the same time, Atlantic inflow to the North Sea weakened to an extent, in both our hindcast and an independent model simulation (Larsen et al., 2016). Partitioning the

influence of Atlantic inflow on North Sea salinity between changes in volume transport and changes in salinity, we may distinguish between "anomalous volume transport of mean salinity" and "mean volume transport of anomalous salinity". Increasing salinity in the Atlantic Water may have thus broadly compensated for declining Atlantic inflow during the 1990s, explaining the absence of an observed salinity trend in the northern North Sea during this period.

Variable Atlantic inflow to the North Sea has a likely impact on North Sea ecosystems via hydrographic changes, as

previously suggested by Reid et al. (2001). The northern North Sea undergoes seasonal stratification, with associated patterns and timings of productivity (Sharples et al., 2006), which may be sensitive to the relative influence of Atlantic Water. Changes of inflow prior to our study period may also help to explain a widely-documented ecosystem regime shift in the early 1980s that was observed in phytoplankton and zooplankton populations (Beaugrand, 2004). Existing EEL observations additionally indicate warming and declining nutrient concentrations in the Rockall trough from 1996 to the

mid-2000s (Johnson et al., 2013), which may have further influenced North Sea ecosystems that previously underwent a regime shift over 1982-88, from a "cold dynamic equilibrium" (1962-83) to a "warm dynamic equilibrium" (1984-99) (Beaugrand, 2004).

Looking back over a longer period, we evaluate sea level differences as proxies for Slope Current transport (since 1957) and Atlantic inflow to the North Sea (since 1965). Slope Current transport variability is identified with sea level differences

between Lerwick (Shetland) and Torshavn (Faroes), while Atlantic inflow to the North Sea is identified with differences between Wick (Scottish mainland) and Lerwick, in both the tide gauge records and in a longer ORCA12 hindcast spanning 1958-2012. In the shorter ORCA12 hindcast that provided the basis for in-depth analysis, Slope Current transport at the Shetland Shelf section is highly correlated with Lerwick-Torshavn sea level differences. Looking to the longer periods, variability of Slope Current transport on a wide range of timescales, from seasonal to multi-decadal, is implicit in Lerwick-

Torshavn sea level differences. Wick-Lerwick sea level differences in tide gauge records indicate considerable decadal variability in the Fair Isle Current transport that dominates Atlantic inflow to the northwest North Sea, while sea level differences in the hindcast are dominated by strong seasonal variability. With locally strong isostacy, differences in local datums, and seasonal steric effects, there are considerable challenges in extracting from tide gauge records the signals that are associated with Slope Current transport and Atlantic inflow. We also recognise that contributions to variability in Lerwick-Torshavn sea level differences may be associated with variability in: (1) the recirculating Faroe Branch of Atlantic inflow; (2) flow that negotiates the Faroe Bank and Wyville Thomson Ridge (Berx et al., 2013). Nevertheless, hindcast Lerwick-Torshavn sea level differences are highly correlated with both the tide gauge equivalent and the hindcast Wick-Lerwick differences. However, the Wick-Lerwick and Lerwick-Torshavn tide gauge differences are not significantly correlated. This suggests that the sea level variability recorded by the tide gauge at Wick is either not capturing the dynamical signal, or is dominated by other influences. Prospects for using sea level records to reconstruct or monitor Atlantic inflow thus depend on refined use of the tide gauge record at Wick.

The larger scale context for long-term changes in the meridional density gradients that support the Slope Current, and Atlantic inflow to the North Sea, likely involves the basin-scale ocean circulation. Previous studies provide evidence for a decline of the Atlantic Meridional Overturning Circulation (AMOC) in mid-latitudes (at 48°N) between the early 1990s and the mid 2000s (Balmaseda et al. 2007; Grist et al. 2009), while Josey et al. (2009) show this decline to be representative across ~48-60°N, a zone encompassing Slope Current inflow. The striking shift to weaker Slope Current transport at the EEL section over 1996-98 coincides with a major warming of the subpolar gyre at this time (Robson et al. 2012). More recently, there has been a major reversal of temperature in the eastern subpolar gyre along with formation of a particularly dense mode of Subpolar Mode Water, associated with extreme cooling in the winter of 2013/14 (Grist et al., 2015), reinforced through further cooling during 2015 (Duchez et al., 2016). These events may have restored strong meridional density gradients and re-strengthened the Slope Current, bringing more high-salinity Atlantic Water to the shelf break. Evidence for such a response in the Slope Current is found in an increase by around 0.05 psu of salinity at around 800 m over much of 2014-16 (see Fig. S11a,b), while the thermal wind relation predicts an approximate doubling of eastward geostrophic transport in mid-latitudes to the west of the shelf break, associated with increased meridional density gradients due to subpolar cooling.

**Author Contribution**

RM designed the study and undertook analysis of the ORCA12-N01 hindcast, including the Lagrangian diagnostics and evaluation with observations (ICES, GODAS). SC analysed the observations of mean absolute dynamic topography and climatological density. MI developed the wind forcing metric. MP analysed drifter observations and calculated the wind forcing metric. IH analysed tide gauge records at Wick, Lerwick and Torshavn. BM diagnosed the ORCA12-N06 hindcast. RM prepared the manuscript with contributions from all co-authors.

## Acknowledgements

RM acknowledges the support of a 2013 Research Bursary awarded by the Scottish Association for Marine Science. MI and MP acknowledge the UK National Environment Research Council (NERC) programme FASTNEt (NERC Ref. NE/I030224/1). SC acknowledges the NERC project UK-OSNAP (NERC Ref. NE/K010700/1) and the EU-funded project NACLIM. BM acknowledges funding from NERC through the RAPID-AMOC Climate Change (RAPID) programme. The ORCA12 simulations were undertaken at the National Oceanography Centre, using the NEMO framework. NEMO is a state-of-the-art, portable modelling framework developed by a consortium of European institutions, namely the National Centre for Scientific Research (CNRS), Paris, the UK Met Office (UKMO), Mercator-Ocean and NERC. GODAS data are provided by the NOAA/OAR/ESRL PSD, Boulder, Colorado, USA, available from their Web site at http://www.esrl.noaa.gov/psd/. We thank two anonymous reviewers for many insightful comments that helped us to substantially improve the manuscript. This study is in memory of Dr Kate Stansfield.

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

Table 1. Long-term mean and standard deviation (SD) of transport, decadal means for 1988-97 and 1998-2007, and corresponding age statistics at the three selected sections along the continental shelf break, and across the Fair Isle Current branch of Atlantic inflow to the northwest North Sea (positive southward). The FASTNEt section extends from 9.74°W, 55.82°N (1516 m) to 9.28°W, 55.79°N (181 m). The EEL section extends from 9.48°W, 57.11°N (1046 m) to 8.57°W, 57.05°N (120 m). The Shetland Slope section extends from 2.72°W, 60.87°N (412 m) to 2.15°W, 60.57°N (133 m). The Fair Isle Current is associated with flow between 1.10°W and 2.46°W at 59.27°N, approximately the western portion of the JONSIS line.

| Section | Transport (Sv) | | | | Travel Time (days) | | | |
| | Mean | SD | Mean | Mean | Since 1 January | | Since 1 July | |
| | | | (88-97) | (98-07) | Mean | SD | Mean | SD |
| --- | --- | --- | --- | --- | --- | --- | --- | --- |
| FASTNEt | 4.62 | 3.34 | 5.94 | 3.30 | 0 | n/a | 0 | n/a |
| EEL | 4.17 | 2.71 | 5.33 | 3.00 | 19.4 | 7.7 | 28.7 | 20.4 |
| Shetland Slope | 7.04 | 2.18 | 7.59 | 6.49 | 90.4 | 23.7 | 98.6 | 32.3 |
| Fair Isle Current | 1.17 | 1.11 | 1.22 | 1.13 | 123.6 | 20.0 | 145.6 | 14.6 |

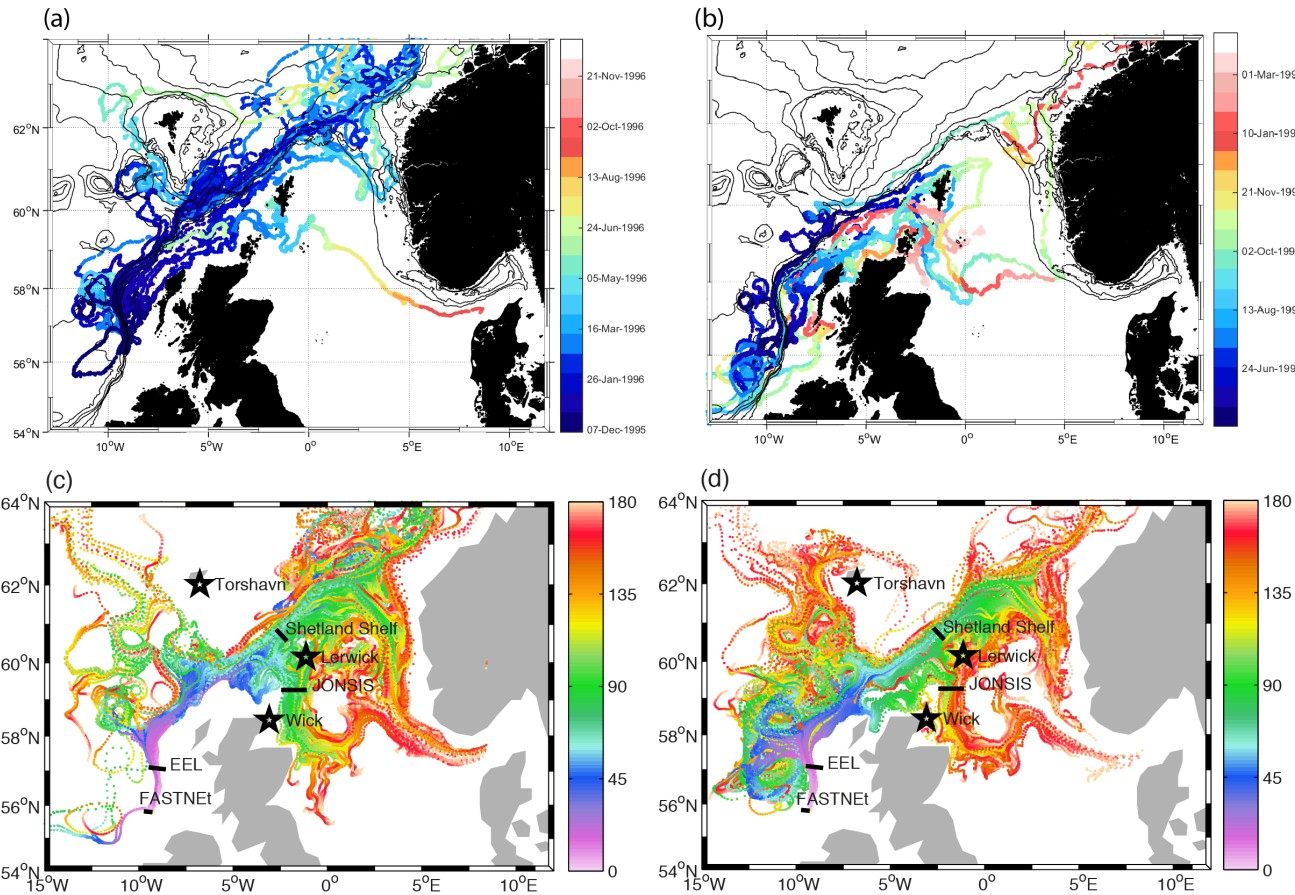

**Figure 1: Drogued drifters released in (a) December 1995, (b) May 1996; model particle trajectories spanning 6 months, released on (c) 1 January 1996, (d) 1 July 1996 (630 model trajectories are plotted in each case). Drifters are colour-coded by calendar date. Model particles are colour-coded by age (days). Note that the drifter data span slightly different durations: up to 11 months, December 1995 – November 1996 (Fig. 1a); up to 10 months, May 1996 – March 1997 (Fig. 1b). In (c) and (d), we also indicate the FASTNEt, EEL and Shetland Shelf sections where we sample the Slope Current, the western JONSIS line where we sample the Fair Isle Current, and the locations of Wick, Lerwick and Torshavn where we take the sea surface height records used to develop proxies for variability of transport in the Slope Current and the Fair Isle Current.**

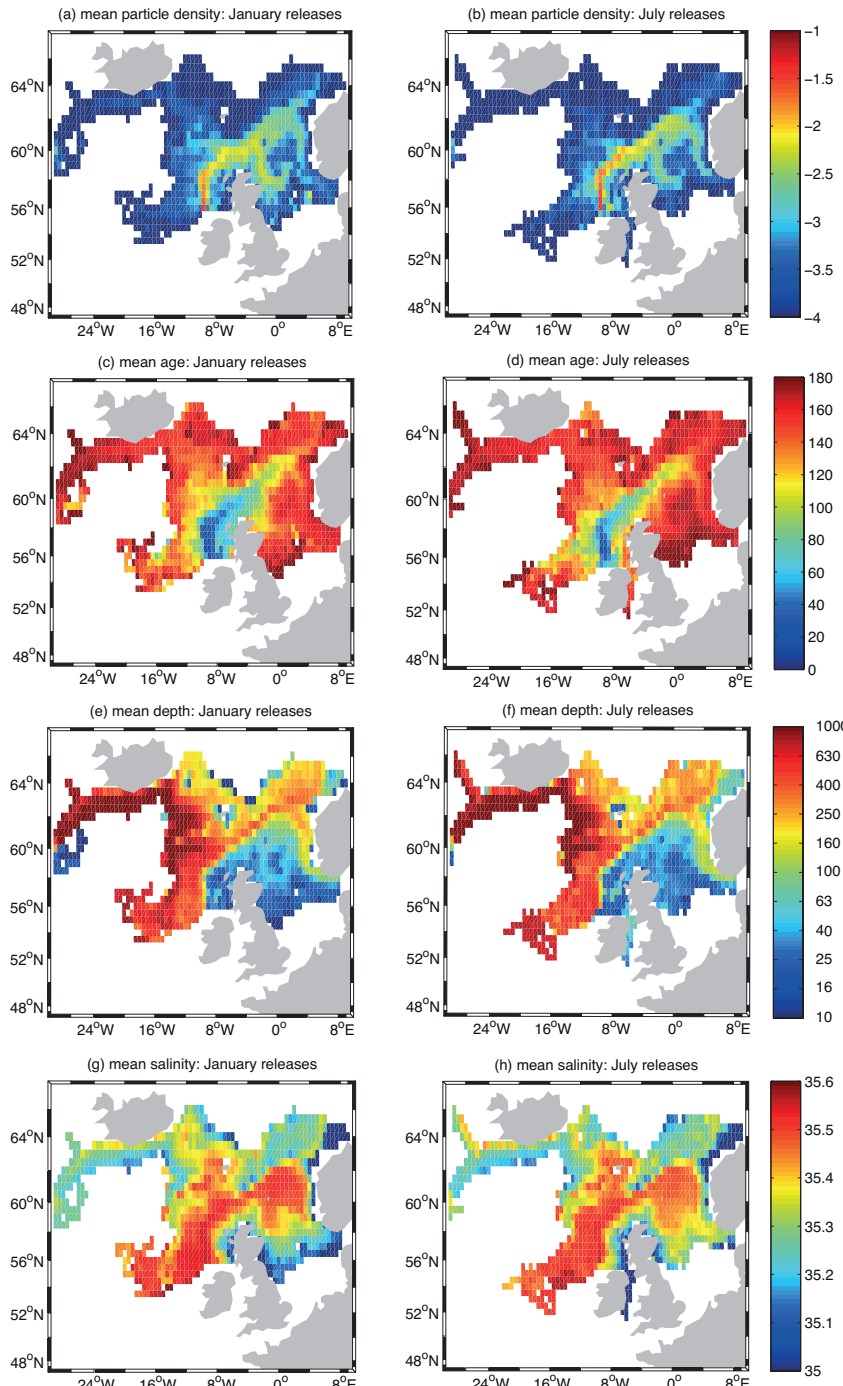

**Figure 2: Forward trajectories from the FASTNEt release section in ORCA12-N01, for particles released on 1 January and 1 July: (a), (b) mean particle density; (c), (d) mean particle age (days since 1 January or 1 July); (e), (f) mean particle depth (m); (g), (h) mean particle salinity (psu). Averages are for 1988-2007, and values are binned at 0.5° x 0.5° resolution. Particle density is expressed as a fraction, obtained as the number of particle occurrences per 0.5° x 0.5° grid cell divided by the total number of particle occurrences. The logarithmic scale for density, ranging from -4 to -1, equates to 0.01-10% of all particle positions.**

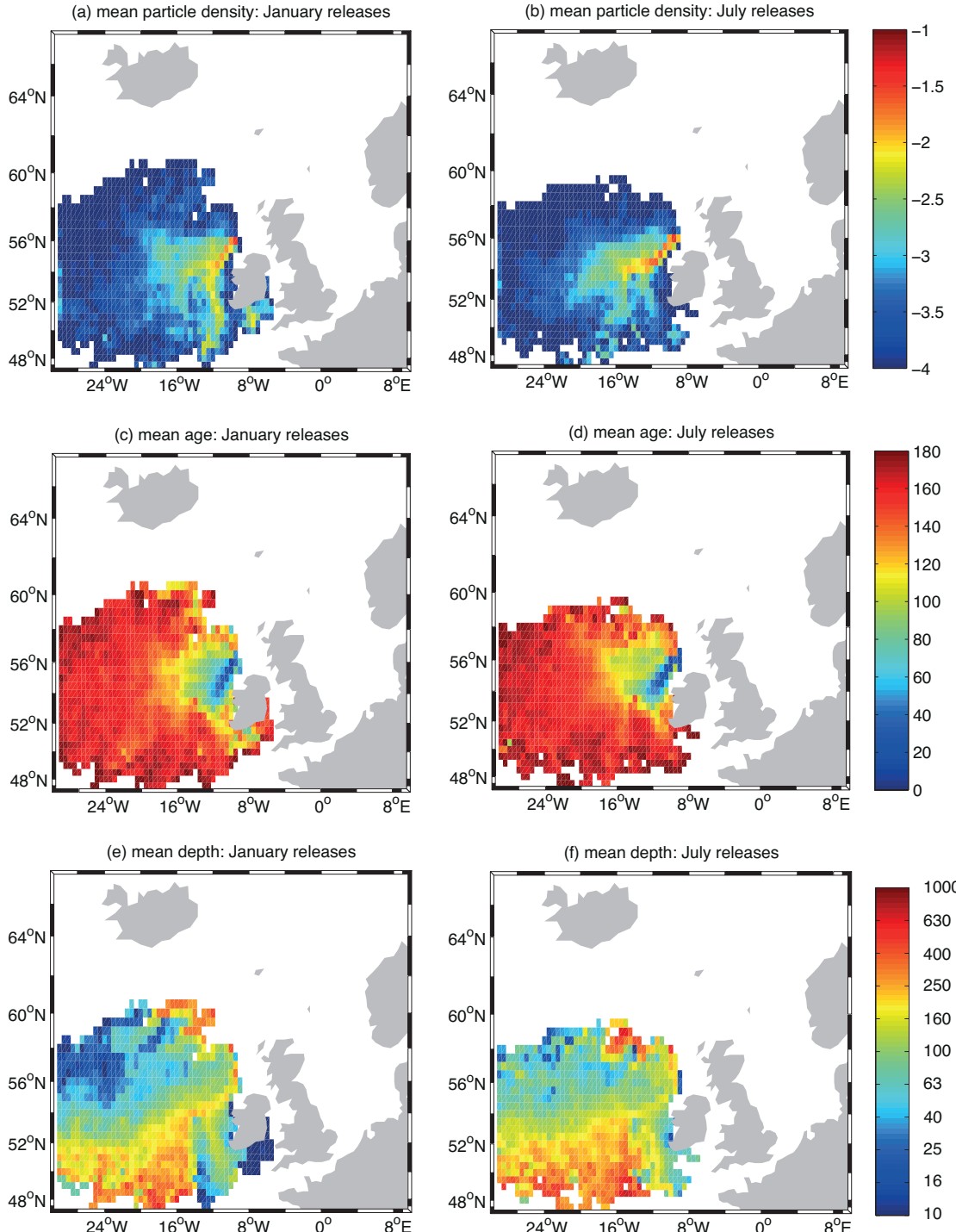

**Figure 3: As Fig. 2a-f, for backward trajectories.**

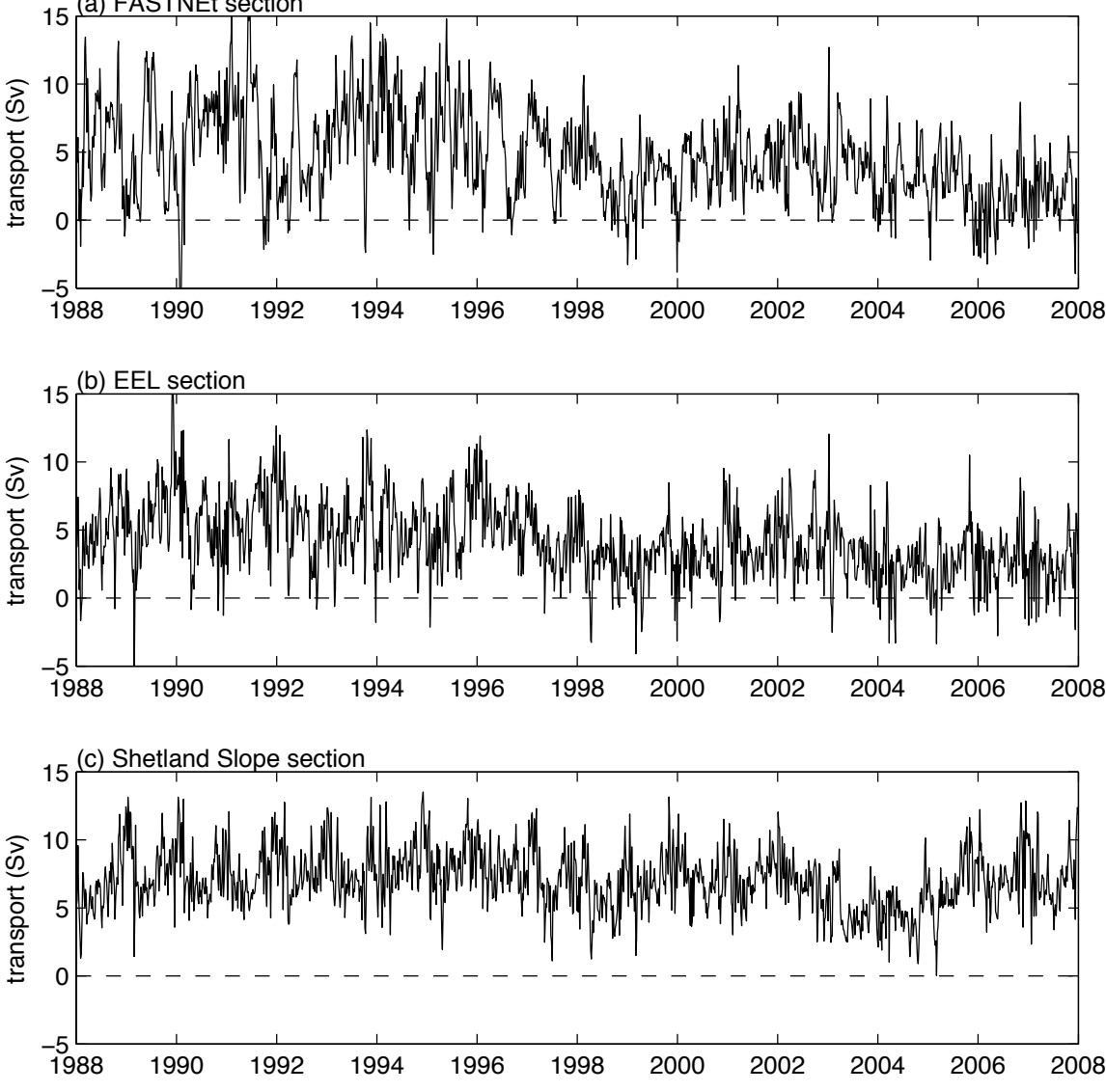

**Figure 4: Slope Current transport at FASTNEt, EEL and Shetland Slope sections, 5-day averaged over 1988-2007 in ORCA12-N01.**

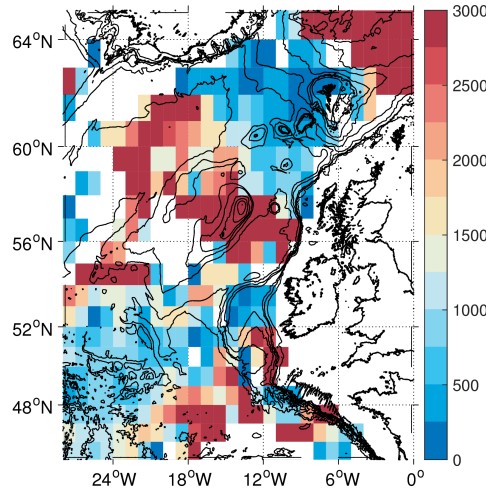

**Figure 5: The depth scale** $H$ **of Eq. (7), predicted from the meridional gradients of density and sea surface height;** $H$ **is set to the local water depth, where** $H$ **exceeds that depth; white areas inside the bold black contour indicate where** $H$ **is negative (undefined).**

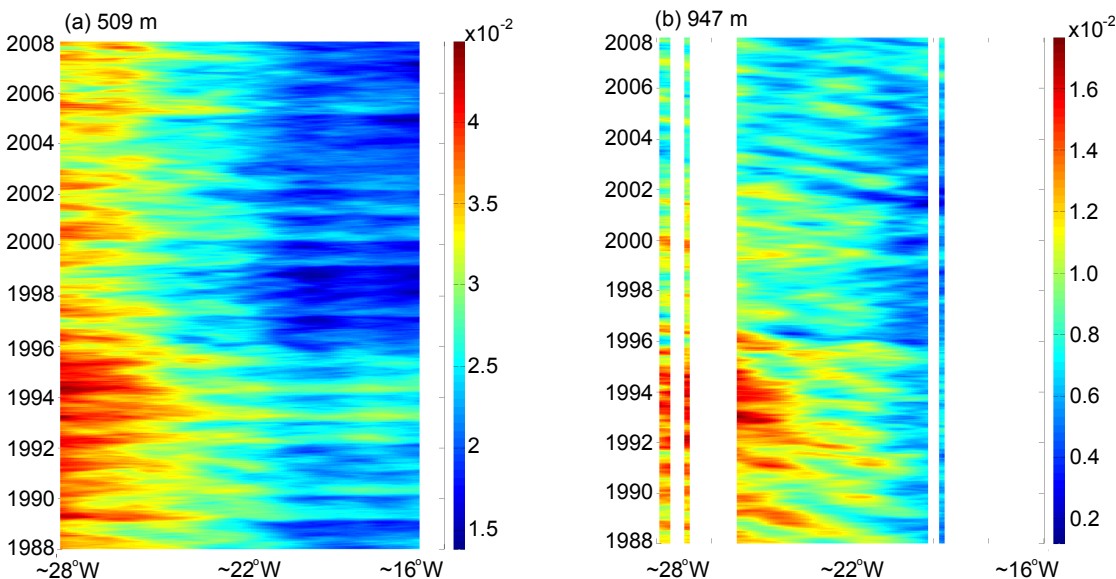

**Figure 6: The northward trend of density ($10^{-2}$ kg m$^{-3}$ degree$^{-1}$, regressed in the latitude range 45-62°N) in the northeast Atlantic (in the approximate longitude range 16-28°W), 5-daily for 1988-2007: (a) at 509 m; (b) at 947 m.**

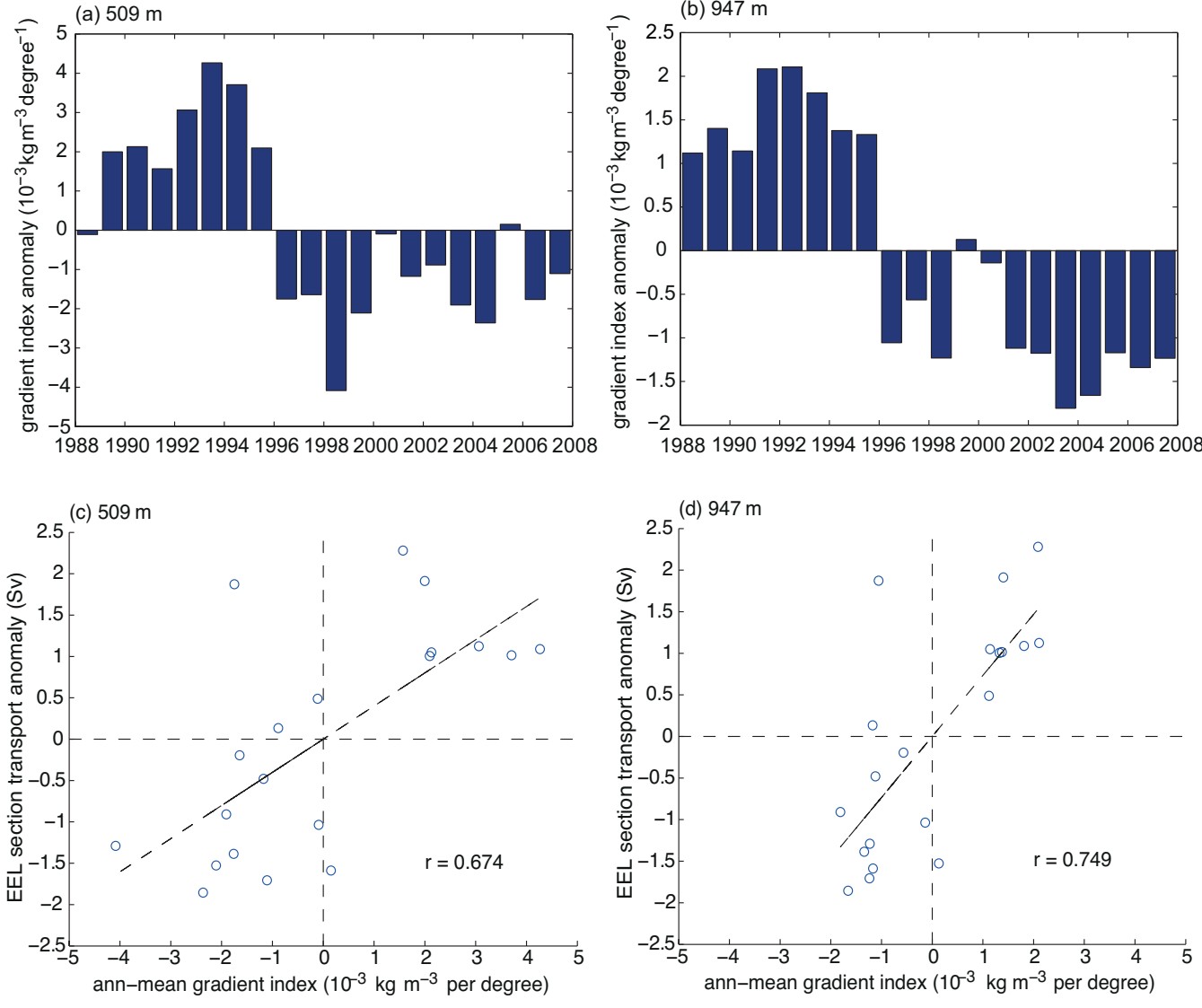

**Figure 7: Time series over 1988-2007, in ORCA12-N01, of anomalies in density gradients averaged across 15-28°W and annually, for 509 m (a) and 947 m (b), and (c), (d), plotted against annual-mean Slope Current (SC) transports at the EEL section.**

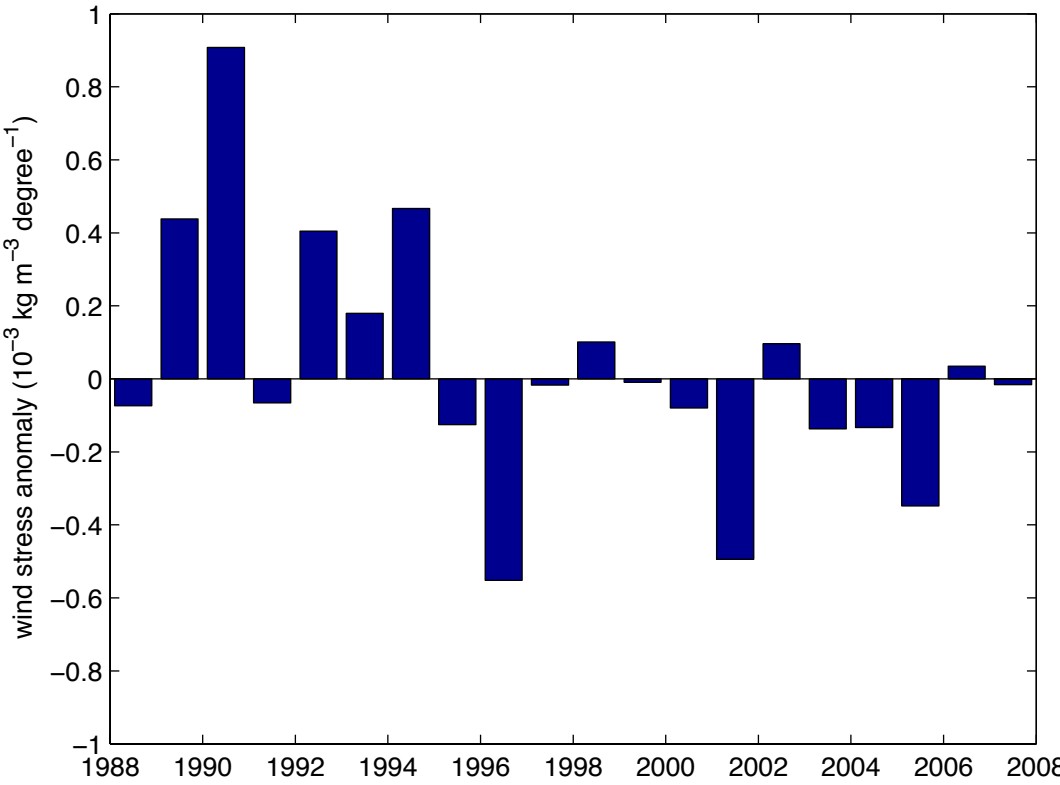

**Figure 8: The wind forcing metric, expressed in units of northward density gradient ($10^{-2}$ kg m$^{-3}$ degree$^{-1}$, regressed in the latitude range 48-62°N) - see text for details.**

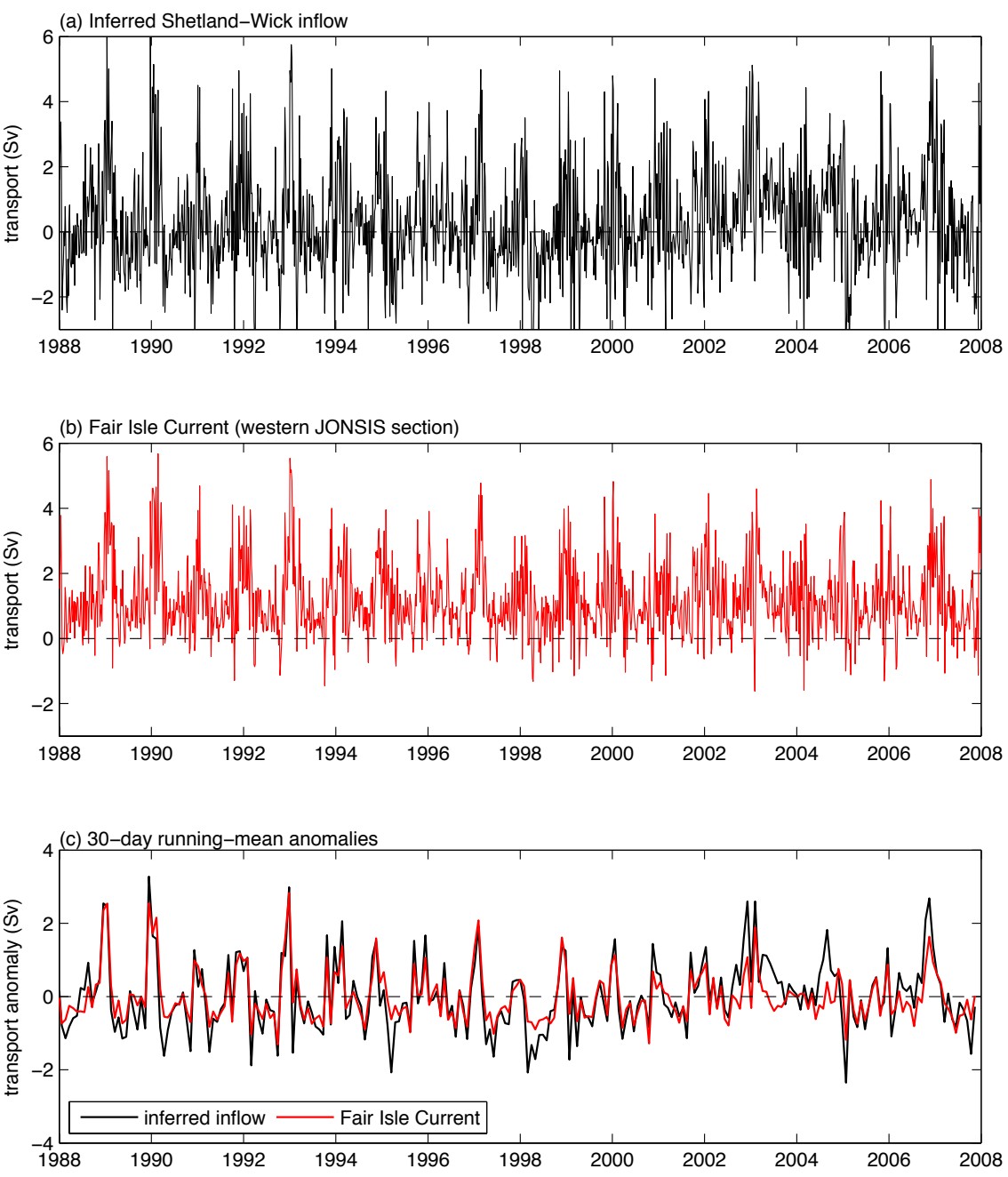

**Figure 9. Transports related to Atlantic inflow into the northwest North Sea: (a) inferred inflow between Shetland and Wick, averaged 5-daily; (b) transport in the Fair Isle Current between 1.10°W and 2.46°W at 59.27°N (along the western JONSIS line), averaged 5-daily; (c) 30-day running mean of anomalies in inferred Atlantic inflow and Fair Isle Current transport.**

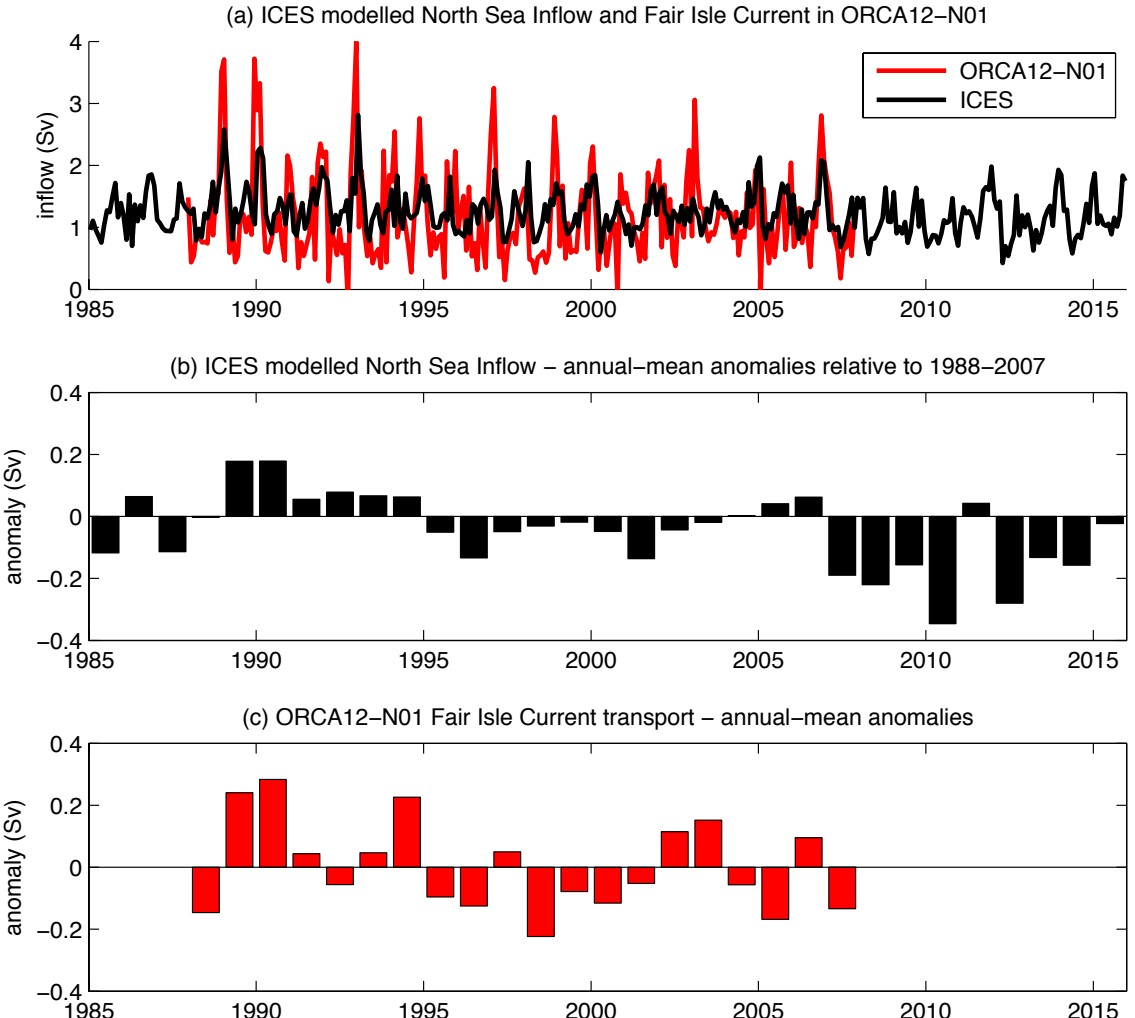

5   **Figure 10: (a) monthly and depth-averaged inflow to North Sea, source: ICES Report on Ocean Climate (thick line), and 30-day running average of 5-day averaged Fair Isle Current transport in ORCA12-N01 (thin line); (b) annual-mean transport anomalies from ICES transport estimates of Atlantic inflow (relative to 1988-2007 mean); (c) annual-mean transport anomalies of Fair Isle Current transport in ORC12-N01.**

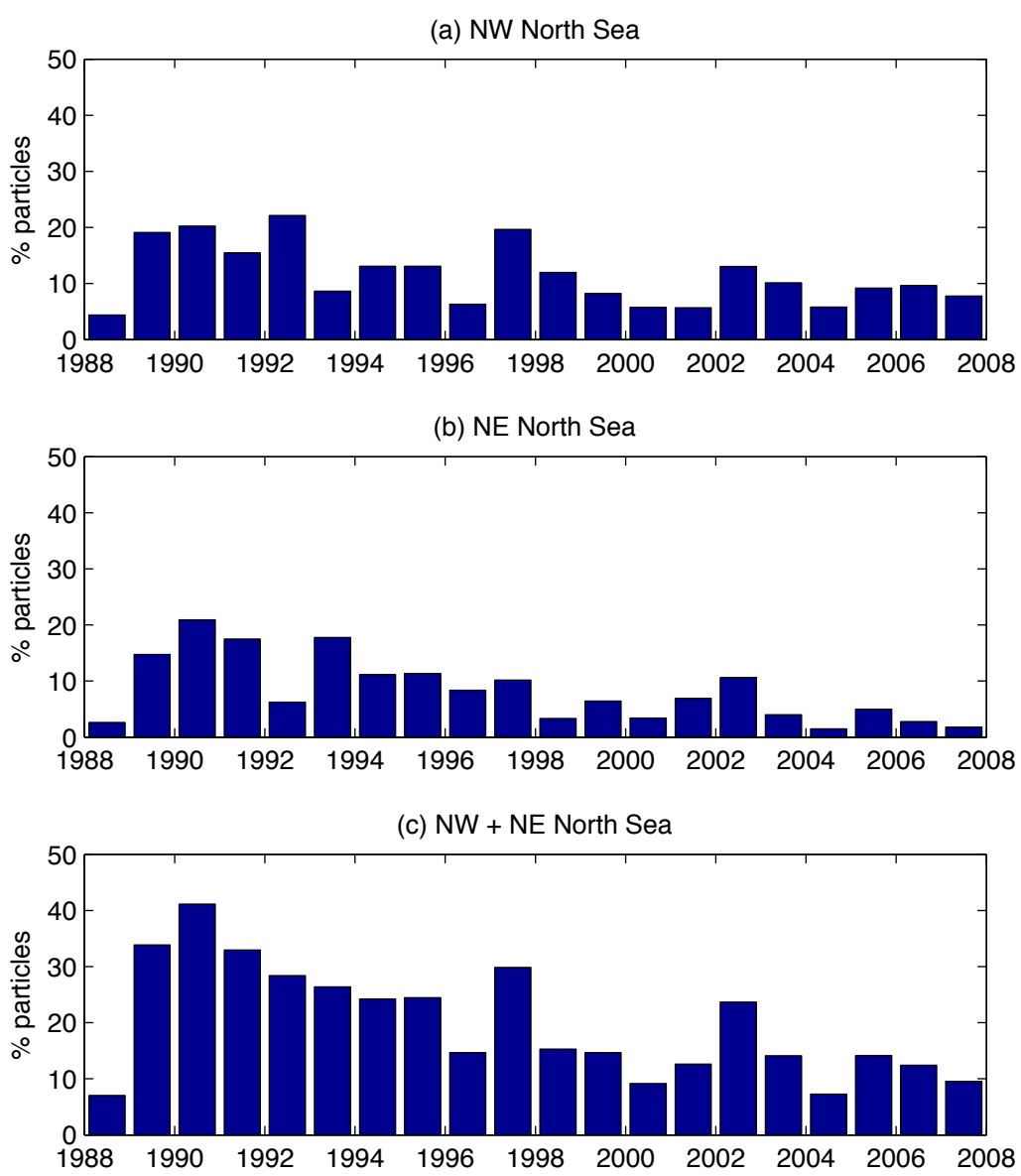

**Figure 11: Histograms (per year) of mean fraction of (combined) January and July released particles residing: (a) in the northwest North Sea (4.5°W to 1.5°E, south of 59°N); (b) in the northeast North Sea (east of 1.5°E, south of 62°N); (c) the combined total.**

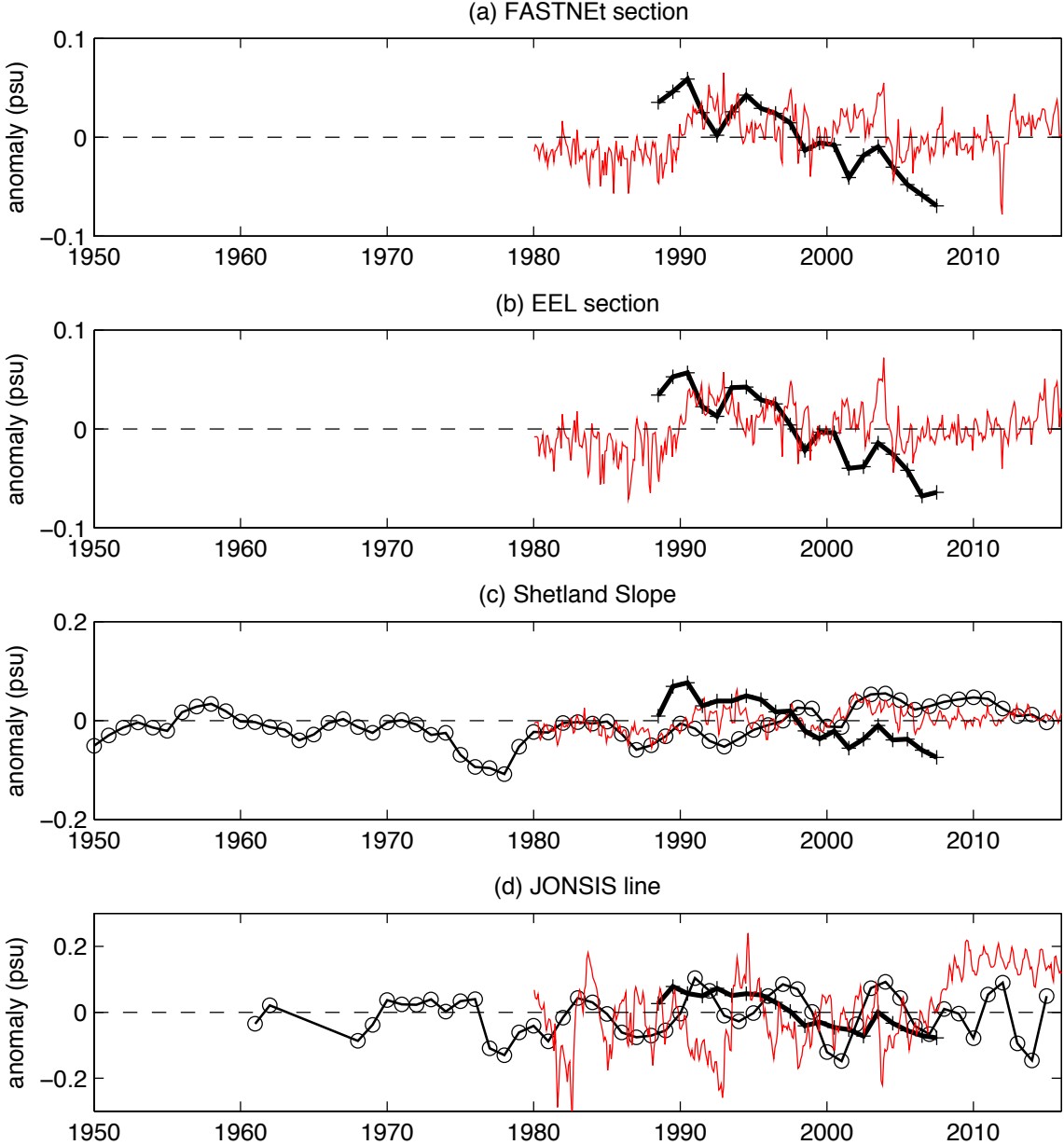

**Figure 12: Annual and monthly mean salinity anomalies (relative to the length of each time series), from observations (thin lines) and in ORCA12-N01 (thick lines), at each section: (a) FASTNEt; (b) EEL; (c) Shetland Slope; (d) JONSIS. The thin red lines are obtained by subtracting climatological seasonal cycles from salinity in the NCEP Global Ocean Data Assimilation System (GODAS) analysis fields, sampled at the following locations: 9.5°W, 55.83°N at 100 m (FASTNEt); 9.5°W, 57.17°N at 100 m (EEL); 2.5°W, 60.83°N at 100 m (Shetland Slope); 1.5°W, 59.17°N at 50 m (JONSIS) – see Figs. S10, S11. The thin black curves in (c), (d) are observations for the Faroe Shetland Channel – Shetland Shelf and the Fair Isle Current (see Sect. 2.1).**

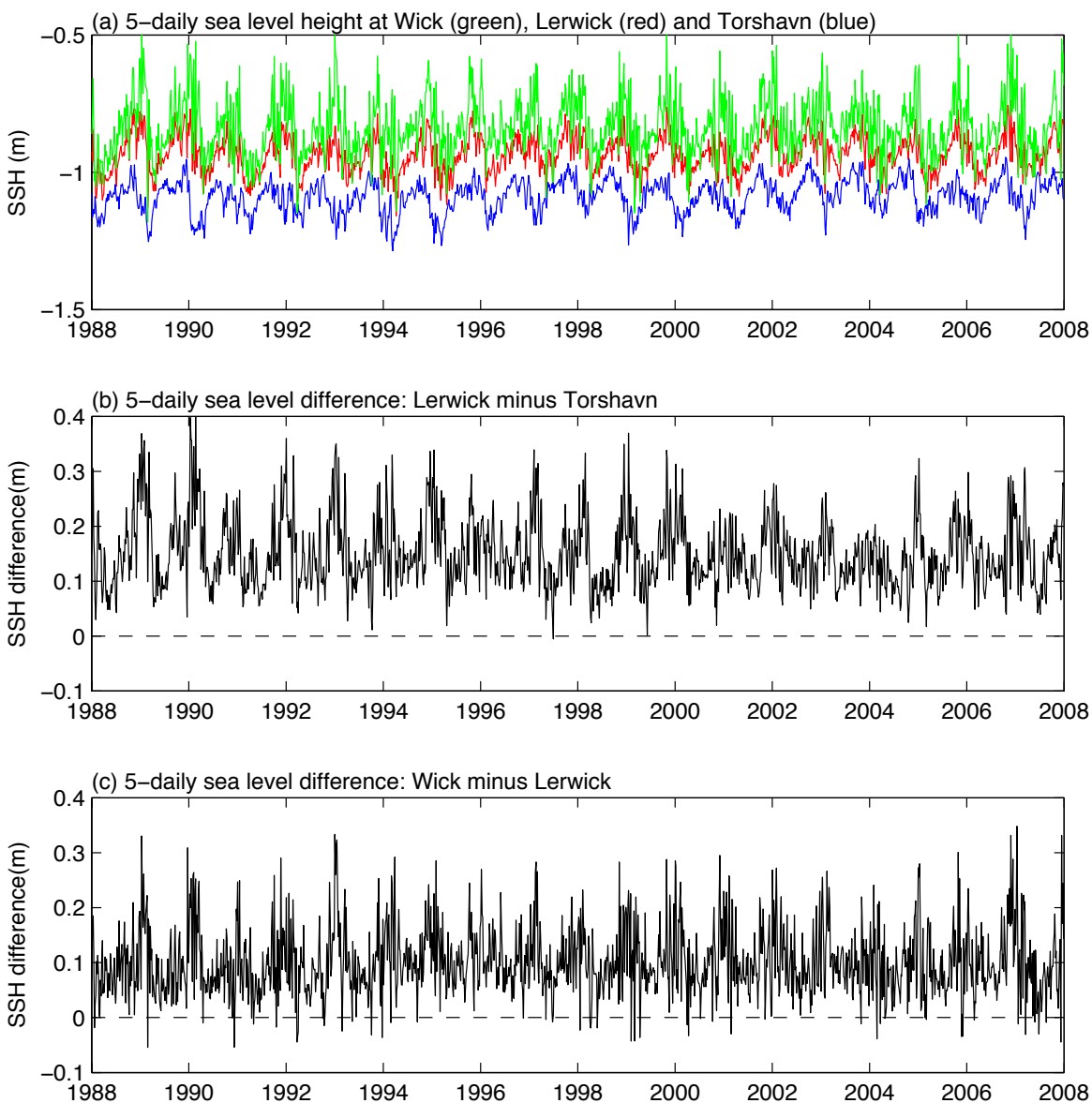

Figure 13: Relative sea surface height at Wick, Lerwick and Torshavn, and the differences, 5-daily averaged over 1988-2007 in ORCA12-N01.

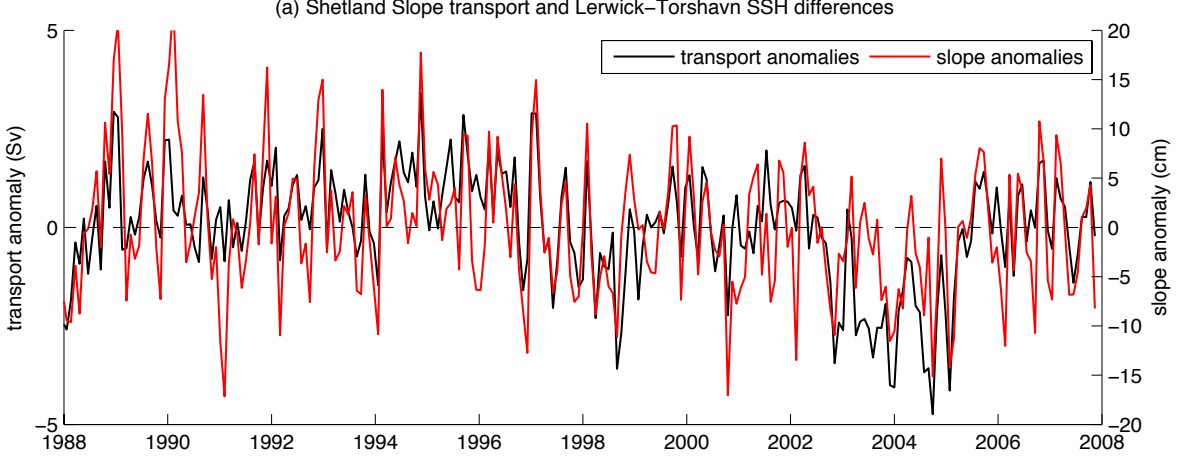

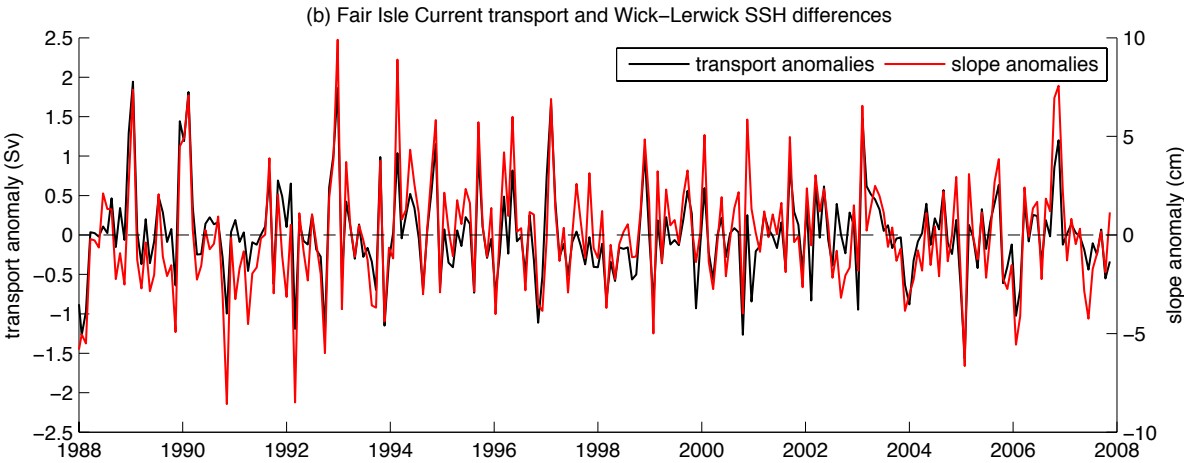

**Figure 14: 30-day running averages of anomalies (relative to seasonal cycles over 1988-2007) of: (a) Shetland Slope transports and Lerwick-Torshavn SSH differences; (b) Fair Isle Current transports and Wick-Lerwick SSH differences.**

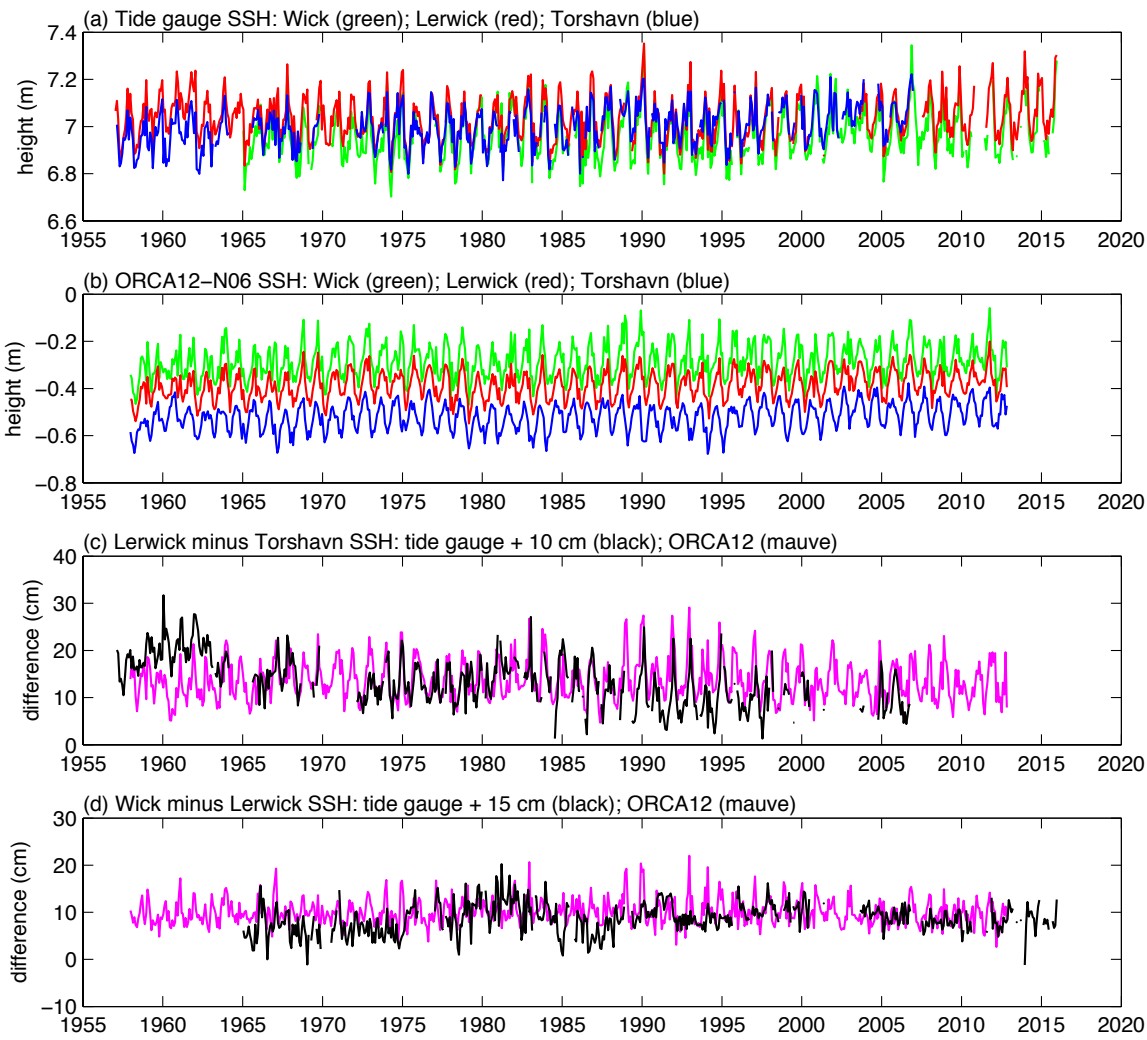

**Figure 15: Sea surface height at Wick, Lerwick and Torshavn: (a) relative sea level (monthly means) from tide gauge records (1957 onwards); (b) relative sea surface height (30-day running means) in ORCA12-N06 (1958-2012); (c) Lerwick minus Torshavn differences; (d) Wick minus Lerwick differences.**

(a) Weak Slope Current

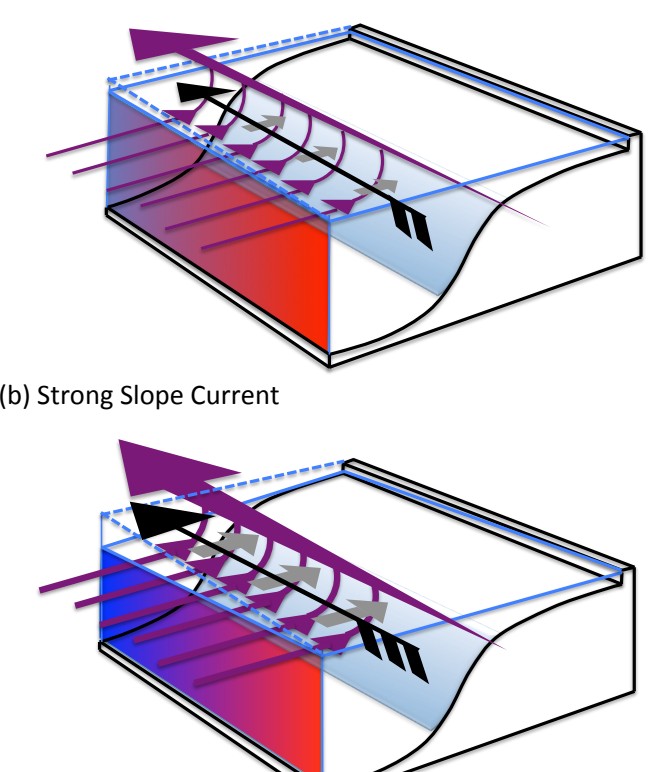

(b) Strong Slope Current

**Figure 16: Schematics showing density gradients (shaded red to pale/dark blue), eastward geostrophic inflow, wind forcing (black alongshore arrow), Ekman transports (grey onshore arrows) and sea surface slopes at an idealized eastern boundary, associated with (a) weak and (b) strong Slope Current transport.**