# Peer review of "Large-scale forcing of the European Slope Current and associated inflows to the North Sea"

_Ocean Science, 2016_

## Referee Comment (RC1) · Anonymous Referee #1 · 26 Sep 2016

General Comments: The manuscript investigates the driving forces of the European Slope Current and also the related inflow of Atlantic water into the north-western North Sea. In this study, a well selected combination of observational data and model results is employed. These data are integrated by means of innovative analytical methods, leading to new insights into the acting mechanisms. The overall impression is that the paper is carefully written in a clear and concise way. I just have some minor comments, which are given below. Therefore, altogether, I could recommend the manuscript for publication after a minor revision.

Detailed Comments: Page 3, line 19: "... are usED" Page 3, Section 2.2, model hindcast: It remains unclear, why ORACA12-N01 data are used at all. If ORACA12-N06 covers a longer period, I do not understand why you do not solely use this data set. Moreover, you must provide more support that a spatial resolution of approx. 10 km

and a temporal resolution of 5 days are sufficient to describe the relevant processes related to the slope current variability. Page 6, line 4: The argument that the number of drifters is limited does not really hold, since it would be possible to start the simulated particles exactly at the same time and place as in the drift experiment. Furthermore, the argument that sub-mesoscale processes hamper a proper comparison between observed and simulated tracers is in contradiction with the statement made in section 2.2, i.e., that eddy-resolving model data are employed. Page 8, line 16: Please clarify how hs and H are defined and give a reasoning why you distinguish between shelf and deep ocean. Actually, the Slope Current, which is in the focus of this study, is located at the transition between these two regions. Which equation holds in these transition areas? Page 8, line 25: Related to the previous comment, I do not see why H is always much larger than hs. Page 12, line 4: The argument here is extremely questionable. If the number of particles entering the North Sea is well correlated with the Slope Current, there is no reason to assume that the definition of the Slope Current is not adequate, when it is correlated with salinity anomalies, since these should also be directly affected by the inflow of Atlantic water. Page 13, line 3: Again, I question why ORCA12-N01 data are used at all, if the ORCA12-N06 data are more reliable as stated here, and cover even a longer period as mentioned earlier. To my opinion, this is unnecessary and just confuses the reader. Page 14, line 23: "Changes OF inflow . . .."

---

## Referee Comment (RC2) · Anonymous Referee #2 · 14 Nov 2016

General Comments

This is a nicely written and well set out paper and I found it easy to follow. The main results from the modelling study have the potential to add to our understanding of the drivers of North Sea inflow. However my main concern is in the analysis of North Sea inflow and the development of the SSH proxy, this aspect of the paper is poorly developed and I don't feel that the conclusions are not well supported. I feel there was limited effort made to validate the model observations, and as a result, that sections 3.4 and 3.5 in particular need some revision.

The use of the SSH metric is puzzling. The authors note that previous researchers have failed to establish a Shetland current transport series based on tide gauge records (Page2, Line 8) but then fail to return to the subject as promised. This statement

should be revised to be more accurate; but more preferable would be a more explicit discussion of this result in Section 4. More importantly there is no mention in this paper of the established pattern of Faroe-Shetland Channel circulation, particularly the recruitment of an additional branch of Atlantic water from the Western Atlantic through the Faroe Bank/Wyville Thomson Ridge region and the recirculation of the North Faroe Current in the channel. Recent estimate of transport through the channel using altimeter data (Berx,2013) demonstrates for example, why a simple SSH metric might not be so useful.

Logically we would expect some relationship between SSH variablity and slope current but it would be worth examining some the different mechanisms for transport of water into the NNS in the NE and the NW for example. This paper could be improved if there was some examination of the relationship between stronger slope current and/or stronger onshelf transport and/or stronger North Sea inflow - the links between these three transports could be the key focus of the paper.

There is no statement in the methods of how the particle backtracking was performed. The first mention of backtracking is in the results section. Also the depth distribution of the tracked particles within the North Sea are not described/discussed. This is of concern because the main Atlantic inflow in NW North Sea is known to flow at depth below the Baltic outflow.

Following on from this, it is not clear why the authors chose to examine surface (only) salinity data within the North Sea (only). Changes in salinity within the North Sea are not just linked to a change in transport of Atlantic Water but also the changing properties of the water that is transported. There was a strong trend of salinification in Atlantic water over the period of the 1990's. This trend is well documented and the reported/observed trend in the NNS is also one of salinification. Having modelled data should offer the authors a chance to investigate these trends and relationships. I believe that doing this would really add value to the paper. Alternatively can the authors explain why they chose surface salinity? Surely the surface salinity metrics are most

likely to show the variability of freshwater flows.

The authors appear surprisingly unaware of the long time-series of data available, both in the northern North Sea and in the Faroe Shetland Channel where there are both hydrographic observations and long term transport estimates. And also the Ellet Line. Before extracting salinity metrics from the North Sea it might be valuable to check how the model represents other observed patterns of salinity

From my understanding of these observations, I am not convinced that there is evidence of a decline in salinity during the 1990's in the North Atlantic-influenced regions of the northern North Sea. Curiously, the modelled decline in salinity almost mirrors the actual increase in salinity observed over the same period in the Slope Current regions and the central North Sea.

I would suggest the following observational evidence be examined.

B. Berx, B. Hansen, S. Østerhus, K. M. Larsen, T. Sherwin, and K. Jochumsen, 'Combining in-Situ Measurements and Altimetry to Estimate Volume, Heat and Salt Transport Variability through the Faroe Shetland Channel', Ocean Sci. Discuss., 10 (2013), 153-95.

Larsen, K. M. H., Gonzalez-Pola, C., Fratantoni, P., Beszczynska-Möller, A., and Hughes, S. L. (Eds). 2016. ICES Report on Ocean Climate 2015. ICES Cooperative Research Report No. 331. 79 pp

N. P. Holliday, S.L. Hughes, S. Bacon, A. Beszczynska-Möller, B. Hansen, A. Lavín, H. Loeng, K.A. Mork, S. Østerhus, T. Sherwin, and W. Walczowski, 'Reversal of the 1960s to 1990s Freshening Trend in the Northeast North Atlantic and Nordic Seas', Geophysical Research Letters, 35 (2008).

S. Dye, N.P. Holliday, S.L. Hughes, M. E. Inall, K. Kennington, T.J. Smyth, J. Tinker, O. Andres, and A. Beszczynska-Moller, 'Impacts of Climate Change on Salinity', in MCCIP Science Review, www.mccip.org.uk/arc, ed. by MCCIP, 2013).

N.P. Holliday, S.L. Hughes, S. Dye, M. E. Inall, J. Read, T. Shammon, T. Sherwin, and T.J. Smyth, 'Salinity in MCCIP Annual Report Card 2010-11', in MCCIP Science Review, 16pp. www.mccip.org.uk/arc, ed. by MCCIP, 2010).

Data from the Feie-Shetland, JONSIS and UTSIRE sections can be obtained from the host institutes or extracted from databases such as BODC, ICES and WODC.

Specific comments:

Page2, Line 11. I'm not convinced that there is "some evidence of surface freshening through the 1990's". Established time series either show no trend in salinity or a slight increase during this period, following the increases seen in Atlantic Water. The freshening that has been observed is limited to the southern North Sea.

Page 7, Line 6-8: The definition of the sections is based on the slope current in the model, but what criteria was applied to define the slope current? See also comments relating to the area of Atlantic water.

Page 7, Line 21 onwards: The authors fail to acknowledge the established knowledge of Faroe-Shetland Channel circulation, particularly the recruitment of an additional branch of Atlantic water from the Western Atlantic through the Faroe Bank/Wyville Thomson Ridge region and the re-circulation of the North Faroe Current in the channel. These offer some explanation as to why a single index of SSH might not represent slope current variability.

Page 11, Line 27. The strong freshening trend from your model data is in the surface salinity of the NW North Sea. Terschelling is a station in the central North Sea and as presented, offers little in support of your observations. The Utsire station as presented by Hjollo sits well within the Baltic Outflow which would link much more strongly to your NE time-series and as mentioned before is likely to mostly reflect variability of freshwater input.

Page 11, Line 34 to Page 12, Line 4: Could the authors elaborate on the spatial variability of what would be their defined Atlantic water mass to highlight what this impact may be?

Page 12, I'm worried about these correlations. All three time-series have a similar downward trend and hence they would always be reasonably correlated. Has this been accounted for in the correlation analysis? If not then by the same argument the wind forcing metric that you include in Figure 8 would also be correlated.

Page 12, Line 4. Have you examined how the salinity of the slope water has increased over the period? Is this reflected in your model? During the periods of weaker modelled transport we know that the slope current became saltier.

Page 12, Lines 15-17: Are the seasonal cycles in the Shetland Slope transport and SSH differences in phase?

Page 13, Line 9. It would be nice to see some comparison of the modelled slope current with some of the observations of the slope current and more sophisticated estimates from altimeter data.

Page 14, Line 11. ref to limited observations. The Northern North Sea is relatively rich in observations, being sampled across two sections (Feie-Shetland and JON-SIS/Utsire) least 5 times per year by UK (Scotland) and Norway. These data are publically available for you to use in your research.

Page 15, Line 6. Sentence starting "These transient events..." In which case presumably we would see a return to higher salinities - has anyone noticed this? I'm not sure you can evidence this statement.

Page 15, Line 13. Reference is also made here to observed trends in the Ellet line data, this raises the question of why the authors did not examine/compare the trends in salinity from the EEL observations as these also shows an increase in salinity during the 1990's?

In methods/supplementary material.

You examine average surface salinity values over a relatively large box areas (NW and NE). The size of these regions makes it likely that both of salinity time-series are capturing variability in Norwegian Coastal current. This is evidenced by the salinity minima in summer in this region. Away from the regions of Baltic influence, in the northern North Sea we would expect salinity maxima in Autumn(September) as a result of Atlantic Inflow. I think it might be valuable to consider different regions and examine data at depth - not just surface values?

Minor/Technical comments

The most common spelling is Faroe and not Faeroe.

Fig 1, Lerwick is not accurately positioned in Figure 1d.The position of the Shetland Slope transect (3) seems to have moved between 1c and 1d.

Fig 1, I find scales hard to read on Figures 1a and 1b, the fontsize used in 1c and 1d is more reasonable.

Fig 2, I note a reference to the log scale - but I think the scale used could be described more clearly.

Fig 5, the number of colours in the scale could align better to the scale intervals.

Page 3, Line 18: ...are used...

Page 6, Line 1. the position of the FASTNEt, EEL and Shetland Shelf sections should be be defined. In section 2.3, floats are mentioned which were deployed for FASTNEt (but which aren't analysed in this manuscript and their position is not described). Is the "FASTNEt section" the line where the ORCA particles were released, if so it needs to be stated on Page 4, Lines 8-10 that this is henceforth referred to as the FASTNEt section. In Table 1 the central locations of EEL and Shetland shelf are mentioned, but the length/endpoints of each section is not defined.

Page 6, line 3. 'northwest' and 'northeast' sectors not adequately defined until 3.4

(Page 11, Line 15) please move that to here or put into methods.

Page 7, line 9. referenced correctly as Sherwin et al 2007. Elsewhere in the paper and in the reference list it is incorrectly noted as Sherwin et al 2008.

Page 7, Line 16. I don't find the more vigorous transport in the early part of the year is 'clearly evident' in this figure. A monthly plot might demonstrate this better.

Page 7, Line 17. Is it possible to quite decadal mean figures rather than relying on our 'general impression'

Page 12, Line 13. Some statistics could be used to determine if this 'impression of generally smaller differences' is correct

Page 13, Line 4. It would be good to know how much smaller - please quote some numbers to back up your statement.

Page 15, Line 9. Please put a date/timescale on 'recent'. It might not be recent when someone reads this next.

Page 15, Line 11. It might be better to say that this study provides "an estimate of the fraction" or "quantifies the fraction".

Page 15, Line 13. starting "Existing EEL observations" These final two sentences feel to me oddly tagged on to the end and of little to do with the conclusion of this paper. If needed, they should be presented within the discussion.

---

## Author Comment (AC1) · 11 Dec 2016

General Comments:

The manuscript investigates the driving forces of the European Slope Current and also the related inflow of Atlantic water into the northwestern North Sea. In this study, a well-selected combination of observational data and model results is employed. These data are integrated by means of innovative analytical methods, leading to new insights into the acting mechanisms. The overall impression is that the paper is carefully written in a clear and concise way. I just have some minor comments, which are given below. Therefore, altogether, I could recommend the manuscript for publication after a minor revision.

Response: We thank the reviewer for supporting and thoughtful comments.

Changes in the manuscript: The manuscript will be revised in accordance with the comments below, alongside more extensive revisions in response to Referee 2.

Detailed Comments:

(1) Page 3, line 19: ". . . are usED"

Response: Noted

Changes in the manuscript: To be corrected

(2) Page 3, Section 2.2, model hindcast: It remains unclear, why ORCA12-N01 data are used at all. If ORACA12-N06 covers a longer period, I do not understand why you do not solely use this data set. Moreover, you must provide more support that a spatial resolution of approx. 10 km and a temporal resolution of 5 days are sufficient to describe the relevant processes related to the slope current variability.

Response: The ORACA12-N06 hindcast was not available until after the long-established ORCA12-N01, and the offline trajectory calculations were until recently only possible with locally archived data (ORACA12-N06 data are archived remotely).

Changes in the manuscript: In Sect. 2.2, we will discuss the extent to which spatial and temporal resolutions are appropriate to a study of the Slope Current dynamics and variability thereof.

(3) Page 6, line 4: The argument that the number of drifters is limited does not really hold, since it would be possible to start the simulated particles exactly at the same time and place as in the drift experiment. Furthermore, the argument that sub-mesoscale processes hamper a proper comparison between observed and simulated tracers is in contradiction with the statement made in section 2.2, i.e., that eddy-resolving model data are employed.

Response: In our statement, we mean that we routinely simulate 630 particle trajec-

tories across a broad depth range representative of the Slope Current (per release), compared to 21 drifters that were drogued to drift with currents at 50 m in the Shelf Edge Study (SES) of LOIS. It would be possible to simulate specifically this number of particle trajectories, with start locations and times as in SES and drifting with currents at the same depth, and we will attempt this. However, the results may not add useful insight, not least because the ocean is chaotic at the mesoscale, so only in statistical terms (i.e., with a larger number of trajectories) are drifts in the ORCA12 hindcast representative of those in the real world. The purpose of Fig. 1 and Sect. 3.1 is to demonstrate that representation of the Slope Current in the ORCA12-N01 hindcast is broadly realistic in terms of pathways and timescales. Regarding sub-mesoscale processes and the point made in Sect. 2.2, we should clarify that ORCA12 resolves only mesoscale processes (10-100 km), but cannot resolve sub-mesoscale processes (1-10 km) such as inertial currents, frontal instabilities, etc.

Changes in the manuscript: In Sect. 3.1, we will clarify the purpose of showing model trajectories alongside drifter tracks, and explain more clearly why we would not expect agreement. We will clarify the distinction between resolved mesoscale and unresolved sub-mesoscale processes. If useful results are obtained with new trajectory calculations (for the same start locations and time as SES, drifting with currents at 50 m), we may show these in place of the current particle trajectories in Fig. 1c,d.

(4) Page 8, line 16: Please clarify how hs and H are defined and give a reasoning why you distinguish between shelf and deep ocean. Actually, the Slope Current, which is in the focus of this study, is located at the transition between these two regions. Which equation holds in these transition areas?

Response: Following Simpson and Sharples (2012), we introduce hs and H in Sect. 3.3.1, following the development of Equation (5) for the meridional sea surface slope in relation to water depth and the meridional density gradient. The reasoning for a distinction between the shelf and deep ocean is in the context of (5), which predicts the scenario presented in Fig. 14, with a steeper (downward to the north) slope in the deep

ocean compared to the shelf, central to our understanding of the Slope Current. Given that the Slope Current is found at a water depth intermediate between hs and H, we would expects an intermediate meridional sea surface slope, but the key issue regards the growing difference between on and off shelf sea surface height with progression to the north, hence strengthening the associated geostrophic flow in the Slope Current.

Changes in the manuscript: We already provide this reasoning for hs and H in the manuscript. On revision, we will review the clarity of this reasoning.

(5) Page 8, line 25: Related to the previous comment, I do not see why H is always much larger than hs.

Response: H is always much larger than hs by definition, as H is the depth of the deep ocean while hs is the depth of the adjacent shelf sea.

Changes in the manuscript: No change necessary

(6) Page 12, line 4: The argument here is extremely questionable. If the number of particles entering the North Sea is well correlated with the Slope Current, there is no reason to assume that the definition of the Slope Current is not adequate, when it is correlated with salinity anomalies, since these should also be directly affected by the inflow of Atlantic water.

Response: Referee 2 raises related points, and we expand on that response here. Contrary to the understanding so far established, we should not neglect variations in salinity, including contributions to Atlantic inflow excluded from our definition of the Slope Current. We can address changing salinity in the model via the salinity along each trajectory (already calculated), and we will further seek observational evidence for any changes.

Changes in the manuscript: We will develop and substantiate statements in Sect. 3.4 that relate to the Slope Current influence on North Sea salinity, due to changes in both transport or salinity, distinguishing between "anomalous volume transport of mean

salinity" and "mean volume transport of anomalous salinity".

(7) Page 13, line 3: Again, I question why ORCA12-N01 data are used at all, if the ORCA12-N06 data are more reliable as stated here, and cover even a longer period as mentioned earlier. To my opinion, this is unnecessary and just confuses the reader.

Response: We do not claim that ORCA12-N06 data are more reliable, only that there is a good agreement in SSH variability between hindcasts over the period of overlap (see Fig. S9). As explained in an earlier response, the ORACA12-N06 hindcast was not available until after the long-established ORCA12-N01, and the offline trajectory calculations were until recently only possible with locally archived data (ORACA12-N06 data are archived remotely). While it may seem desirable to repeat the trajectory calculations with ORACA12-N06, these involve considerable effort and would severely delay manuscript revisions, while not substantively changing the conclusions of this study. We anticipate further studies of Slope Current variability that will involve the use of new and longer hindcasts. We chose to use ORCA12-N06 for the analysis in Sect. 3.5, as it is relatively straightforward to obtain model SSH data for direct comparison with the tide gauge data over the longer period.

Changes in the manuscript: At the end of Sect. 2.2, we will further clarify why we chose to use the new ORCA12 hindcast.

(8) Page 14, line 23: "Changes OF inflow ...."

Response: Noted

Changes in the manuscript: To be corrected

---

## Author Comment (AC2) · 11 Dec 2016

General Comments:

This is a nicely written and well set out paper and I found it easy to follow. The main results from the modelling study have the potential to add to our understanding of the drivers of North Sea inflow. However my main concern is in the analysis of North Sea inflow and the development of the SSH proxy, this aspect of the paper is poorly developed and I don't feel that the conclusions are not well supported. I feel there was limited effort made to validate the model observations, and as a result, that sections 3.4 and 3.5 in particular need some revision.

[Figure]

Response: We thank the reviewer for broad support and a very thorough review.

Changes in the manuscript: The manuscript will be extensively revised, paying particular attention to Sections 3.2, 3.4 and 3.5.

(1) The use of the SSH metric is puzzling. The authors note that previous researchers have failed to establish a Shetland current transport series based on tide gauge records (Page 2, Line 8) but then fail to return to the subject as promised. This statement should be revised to be more accurate; but more preferable would be a more explicit discussion of this result in Section 4. More importantly there is no mention in this paper of the established pattern of Faroe-Shetland Channel circulation, particularly the recruitment of an additional branch of Atlantic water from the Western Atlantic through the Faroe Bank/Wyville Thomson Ridge region and the recirculation of the North Faroe Current in the channel. Recent estimate of transport through the channel using altimeter data (Berx et al., 2013) demonstrates for example, why a simple SSH metric might not be so useful.

Response: We appreciate this comment, and agree that further discussion and clarification is necessary.

Changes in the manuscript: We will extend the Introduction, selected parts of the Results, and the Discussion accordingly.

(2) Logically we would expect some relationship between SSH variability and slope current but it would be worth examining some the different mechanisms for transport of water into the NNS in the NE and the NW for example. This paper could be improved if there was some examination of the relationship between stronger slope current and/or stronger onshelf transport and/or stronger North Sea inflow - the links between these three transports could be the key focus of the paper.

Response: We agree that further analysis may provide helpful insight as to the relative contributions of Slope Current (as currently defined – see Sect. 3.2) and on-shelf

transport to North Sea inflow.

Changes in the manuscript: We will diagnose transport inshore of the Slope Current at the "Shetland Slope section", and compare this to Slope Current transport. We will further diagnose North Sea inflow at the Feie-Shetland and Jonis-Utsire sections. Pending the results of this new analysis, we will revise Sect. 3.4 and the Discussion.

(3) There is no statement in the methods of how the particle backtracking was performed. The first mention of backtracking is in the results section. Also the depth distribution of the tracked particles within the North Sea is not described/discussed. This is of concern because the main Atlantic inflow in NW North Sea is known to flow at depth below the Baltic outflow.

Response: ARIANE (as other tracking methods) can be used in backward mode, as forward mode, simply by time-reversing the analytical calculation of particle progress through grid-cells. Depth per trajectory is recorded. Depth distributions within the North Sea will be examined, to establish whether indeed inflow in the NW North Sea in ORCA12 is found below the fresh Baltic outflow.

Changes in the manuscript: This will be explained in Sect. 2.3. North Sea particle depth distributions will be plotted as for particle density and age. Figures will be shown in either the main section or as Supplementary Material. Sect. 3.2 will be revised accordingly.

(4) Following on from this, it is not clear why the authors chose to examine surface (only) salinity data within the North Sea (only). Changes in salinity within the North Sea are not just linked to a change in transport of Atlantic Water but also the changing properties of the water that is transported. There was a strong trend of salinification in Atlantic water over the period of the 1990's. This trend is well documented and the reported/observed trend in the NNS is also one of salinification. Having modelled data should offer the authors a chance to investigate these trends and relationships. I believe that doing this would really add value to the paper. Alternatively can the authors

explain why they chose surface salinity? Surely the surface salinity metrics are most likely to show the variability of freshwater flows.

Response: Our choice to analyse only surface salinity within the North Sea is admittedly limiting, and we agree that a more analysis is warranted here. Indeed, both v' (changes in transport) and S' (changes in property) should be considered.

Changes in the manuscript: We will undertake a wider analysis of salinity variability, across the study region and for different depths, also informed by the depths followed by particles through the North Sea region in particular. Rather than over-lengthen the manuscript with an entirely new section, we will incorporate an expanded salinity analysis into Sect. 3.4, with additional figures added to Supplementary Material.

(5) The authors appear surprisingly unaware of the long time-series of data available, both in the northern North Sea and in the Faroe Shetland Channel where there are both hydrographic observations and long-term transport estimates. And also the Ellett Line. Before extracting salinity metrics from the North Sea it might be valuable to check how the model represents other observed patterns of salinity.

Response: Thank you for pointing out the available observations. Our awareness of this data is implicit in citation of Sherwin et al. (2008), but we will indeed look to model validation alongside observed transport estimates and salinity variations, as published. We have already obtained relevant data provided online with the ICES Report on Ocean Climate (http://ocean.ices.dk/iroc/).

Changes in the manuscript: If and where practical, we will co-plot observed and model estimates. Otherwise, we will discuss model transports and variability in the context of published observations. We will revise Sect. 3.2 and 3.4 accordingly.

(6) From my understanding of these observations, I am not convinced that there is evidence of a decline in salinity during the 1990's in the North Atlantic-influenced regions of the northern North Sea. Curiously, the modelled decline in salinity almost mirrors the

actual increase in salinity observed over the same period in the Slope Current regions and the central North Sea.

Response: Clearly, this is an important matter for clarification. Once we have addressed the broader 3D changes in salinity, and compared these with observations, it will be evident whether model is reliably simulating observed changes in the Slope Current system, both transports and properties. We can then re-visit statements about North Sea salinity and links with variable Slope Current transport.

Changes in the manuscript: In line with previous outlined changes, Sect. 3.4 will be revised accordingly.

(7) I would suggest the following observational evidence be examined.

B. Berx, B. Hansen, S. Østerhus, K. M. Larsen, T. Sherwin, and K. Jochumsen, 'Combining in-Situ Measurements and Altimetry to Estimate Volume, Heat and Salt Transport Variability through the Faroe Shetland Channel', Ocean Sci. Discuss., 10 (2013), 153-95.

Larsen, K. M. H., Gonzalez-Pola, C., Fratantoni, P., Beszczynska-Möller, A., and Hughes, S. L. (Eds). 2016. ICES Report on Ocean Climate 2015. ICES Cooperative Research Report No. 331. 79 pp

N. P. Holliday, S.L. Hughes, S. Bacon, A. Beszczynska-Möller, B. Hansen, A. Lavín, H. Loeng, K.A. Mork, S. Østerhus, T. Sherwin, and W. Walczowski, 'Reversal of the 1960s to 1990s Freshening Trend in the Northeast North Atlantic and Nordic Seas', Geophysical Research Letters, 35 (2008).

S. Dye, N.P. Holliday, S.L. Hughes, M. E. Inall, K. Kennington, T.J. Smyth, J. Tinker, O. Andres, and A. Beszczynska-Moller, 'Impacts of Climate Change on Salinity', in MCCIP Science Review, www.mccip.org.uk/arc, ed. by MCCIP, 2013).

N.P. Holliday, S.L. Hughes, S. Dye, M. E. Inall, J. Read, T. Shammon, T. Sherwin, and T.J. Smyth, 'Salinity in MCCIP Annual Report Card 2010-11', in MCCIP Science

Review, 16pp. www.mccip.org.uk/arc, ed. by MCCIP, 2010).

Data from the Feie-Shetland, JONSIS and UTSIRE sections can be obtained from the host institutes or extracted from databases such as BODC, ICES and WODC.

Response: Thank you for drawing these publications and data to our attention.

Changes in the manuscript: Relevant publications will be cited, and selected observations will be sought and used to evaluate the simulation.

Specific comments:

(8) Page 2, Line 11. I'm not convinced that there is "some evidence of surface freshening through the 1990's". Established time series either show no trend in salinity or a slight increase during this period, following the increases seen in Atlantic Water. The freshening that has been observed is limited to the southern North Sea.

Response: This statement was based on inspection of figures in Højllo et al. (2009), as indicated. We accept the referee's point about different salinity changes elsewhere, and we will re-visit the broader patterns and character of salinity change.

Changes in the manuscript: The Introduction will be revised accordingly, with reference to further observations, outlining the pattern and character of salinity changes across the North Sea.

(9) Page 7, Line 6-8: The definition of the sections is based on the slope current in the model, but what criteria was applied to define the slope current? See also comments relating to the area of Atlantic water.

Response: The Slope Current was identified as a narrow band of high velocity current confined between isobaths, in 5-day averaged ORCA12 datasets.

Changes in the manuscript: In Sect. 3.2, we will specify the criteria for locating the Slope Current, both for calculating transport and for specifying initial particle locations.

(10) Page 7, Line 21 onwards: The authors fail to acknowledge the established knowledge of Faroe-Shetland Channel circulation, particularly the recruitment of an additional branch of Atlantic water from the Western Atlantic through the Faroe Bank/Wyville Thomson Ridge region and the re-circulation of the North Faroe Current in the channel. These offer some explanation as to why a single index of SSH might not represent slope current variability.

Response: Thank you for raising this point, which we should certainly bear in mind, in regard to the drivers of variability in a SSH slope index. From the ORCA12 simulation, We will extract SSH differences across just the Slope Current, for comparison with SSH differences between Lerwick and Torshavn. The strength of correlation between time series of these two indices will inform this issue.

Changes in the manuscript: We will revise Sect. 3.5 accordingly, with the utility of a Lerwick-Torshavn SSH index for Slope Current transport more or less supported by the correlation between this index and SSH slope across just the Slope Current.

(11) Page 11, Line 27. The strong freshening trend from your model data is in the surface salinity of the NW North Sea. Terschelling is a station in the central North Sea and as presented, offers little in support of your observations. The Utsire station as presented by Hjollo sits well within the Baltic Outflow which would link much more strongly to your NE time-series and as mentioned before is likely to mostly reflect variability of freshwater input.

Response: Thank you for pointing this out. We will indeed look more widely to the available salinity observations and in particular those at locations/depths sensitive to variations in Atlantic inflow to the North Sea.

Changes in the manuscript: Pending further analysis of salinity observations, we will revise Sect. 3.4 accordingly.

(12) Page 11, Line 34 to Page 12, Line 4: Could the authors elaborate on the spatial

variability of what would be their defined Atlantic water mass to highlight what this impact may be?

Response: We can further analyse the spatial distribution of salinity in the ORCA12 hindcast to substantiate this remark. The principal water mass in the Faroe-Shetland slope is North Atlantic Water (McKenna et al., 2016), but we will look to the wider literature in regard to on-shelf water masses.

Changes in the manuscript: Pending further model analysis and a review of relevant literature, we will provide clarification accordingly.

(13) Page 12, I'm worried about these correlations. All three time-series have a similar downward trend and hence they would always be reasonably correlated. Has this been accounted for in the correlation analysis? If not then by the same argument the wind forcing metric that you include in Figure 8 would also be correlated.

Response: Trends are included in the correlated data. We agree that there is limited value in correlations between short time series (given only 20 annual values) that are characterized by similar trends.

Changes in the manuscript: We may omit correlations from Sect. 3.4 and Sect. 3.5 of the revised manuscript.

(14) Page 12, Line 4. Have you examined how the salinity of the slope water has increased over the period? Is this reflected in your model? During the periods of weaker modelled transport we know that the slope current became saltier.

Response: We can address changing salinity in the model via the salinity along each trajectory (already calculated), and we will further seek observational evidence for any changes.

Changes in the manuscript: We will develop and substantiate statements in Sect. 3.4 that relate to the Slope Current influence on North Sea salinity, due to changes in both transport or salinity, distinguishing between "anomalous volume transport of mean

salinity" and "mean volume transport of anomalous salinity".

(15) Page 12, Lines 15-17: Are the seasonal cycles in the Shetland Slope transport and SSH differences in phase?

Response: Yes, Shetland Slope transport and the Lerwick-Torshavn SSH difference are in phase.

Changes in the manuscript: We will emphasize this in-phase relationship in Sect. 3.5.

(16) Page 13, Line 9. It would be nice to see some comparison of the modelled slope current with some of the observations of the slope current and more sophisticated estimates from altimeter data.

Response: We are aware of transport estimates for the Atlantic inflow between the Faroes and Shetland since December 1992 (Fig. 10 in Berx et al., 2013) in the range 1-5 Sv – similar to the range seen in ORCA12 (our Fig. 4c).

Changes in the manuscript: We will obtain ORCA12-N01 transports for the full Faroes-Shetland section (as Berx et al., 2013), and present this time series as an additional panel (Fig. 4d) below that for our "Shetland Slope" transport estimate (Fig. 4c). We will thus directly compare ORCA12 transports with corresponding observations, and also determine the extent to which Slope Current variability on the Shetland Slope accounts for variability across the full section.

(17) Page 14, Line 11. ref to limited observations. The Northern North Sea is relatively rich in observations, being sampled across two sections (Feie-Shetland and JON- SIS/Utsire) least 5 times per year by UK (Scotland) and Norway. These data are publically available for you to use in your research.

Response: We will access the necessary data, as recommended.

Changes in the manuscript: We will refer to salinity variations at the Feie-Shetland and JON- SIS/Utsire sections, and revise our conclusions accordingly.
(18) Page 15, Line 6. Sentence starting "These transient events..." In which case presumably we would see a return to higher salinities - has anyone noticed this? I'm not sure you can evidence this statement.

Response: This is a good point. We are currently examining the GODAS ocean analysis (http://www.esrl.noaa.gov/psd/data/gridded/data.godas.html) for evidence of recent changes in the Slope Current and regional salinity that are consistent with strengthened meridional density gradients and geostrophic inflow.

Changes in the manuscript: We will revise the discussion in accordance with our findings and any other evidence.

(19) Page 15, Line 13. Reference is also made here to observed trends in the Ellett line data, this raises the question of why the authors did not examine/compare the trends in salinity from the EEL observations as these also shows an increase in salinity during the 1990's?

Response: We will access the EEL salinity observations, specifically in the Slope Current, as recommended. Repeating an earlier response, we can address changing salinity in the model via the salinity along each trajectory (already calculated), which we can compare to the EEL data. Following Fig. S1 and Fig. S2, we will plot mean particle salinity for January and July releases of each year. At the EEL section, we can plot this salinity as a time series, for comparison with observations.

Changes in the manuscript: We will include further plots in Supplementary Material, as outlined above, refer to these in Sect. 3.2. In Sect. 4, we will discuss observed and simulated salinity changes, over 1988-2007, in the Slope Current to the west of Scotland.

(20) In methods/supplementary material: You examine average surface salinity values over relatively large box areas (NW and NE). The size of these regions makes it likely that both of salinity time-series are capturing variability in Norwegian Coastal current.

[Figure]

This is evidenced by the salinity minima in summer in this region. Away from the regions of Baltic influence, in the northern North Sea we would expect salinity maxima in Autumn (September) as a result of Atlantic Inflow. I think it might be valuable to consider different regions and examine data at depth - not just surface values?

Response: Thank you for this insight. We will re-calculate salinity, this time also for full-depth (rather than just the surface), for re-defined, perhaps irregular, sub-regions that are informed by the pattern of inflow evident in Fig. 2.

Changes in the manuscript: We will replace Fig. 10 with time series of salinity anomalies for the redefined sub-regions, possibly with additional panels relating to the salinity anomalies in the upstream Slope Current (see previous responses). We will revise the text in Sect. 3.4 accordingly.

Minor/Technical comments:

(21) The most common spelling is Faroe and not Faeroe.

Response: Accepted

Changes in the manuscript: Text will be revised accordingly.

(22) Fig 1, Lerwick is not accurately positioned in Figure 1d. The position of the Shetland Slope transect (3) seems to have moved between 1c and 1d.

Response: Accepted

Changes in the manuscript: Fig. 1c,d will be revised accordingly.

(23) Fig 1, I find scales hard to read on Figures 1a and 1b, the font size used in 1c and 1d is more reasonable.

Response: Accepted

Changes in the manuscript: Fig. 1a,b will be revised accordingly.

(24) Fig 2, I note a reference to the log scale - but I think the scale used could be

described more clearly.

Response: Agreed

Changes in the manuscript: The purpose of "particle density" shown in Figs. 2a,b and Fig. 3a,b is to reveal dominant pathways through the region. We will elaborate in the caption that the density diagnostic records particle presence per grid cell in the range 0.01-10% (of all particle positions).

(25) Fig 5, the number of colours in the scale could align better to the scale intervals.

Response: Point taken

Changes in the manuscript: We will re-plot the figure accordingly.

(26) Page 3, Line 18: ...are used...

Response: Noted (as Referee 1)

Changes in the manuscript: Text will be corrected

(27) Page 6, Line 1. the position of the FASTNEt, EEL and Shetland Shelf sections should be be defined. In section 2.3, floats are mentioned which were deployed for FASTNEt (but which aren't analysed in this manuscript and their position is not described). Is the "FASTNEt section" the line where the ORCA particles were released, if so it needs to be stated on Page 4, Lines 8-10 that this is henceforth referred to as the FASTNEt section. In Table 1 the central locations of EEL and Shetland shelf are mentioned, but the length/endpoints of each section is not defined.

Response: The position of the "FASTNEt section" is indeed as defined in Sect. 2.3. We agree that full details of the EEL and Shetland Shelf sections should be defined.

Changes in the manuscript: We will make clear in Sect. 2.3 that we are defining the FASTNEt section, from which particles are tracked forwards and backwards. The endpoints for EEL and Shetland Shelf sections will be provided in Sect. 3.2 and in the
caption for Table 1.

(28) Page 6, line 3. 'northwest' and 'northeast' sectors not adequately defined until 3.4 (Page 11, Line 15) please move that to here or put into methods.

Response: This is an oversight.

Changes in the manuscript: We will define northwest and northeast sectors of the North Sea in Sect. 2.3, where we introduce the Lagrangian methodology.

(29) Page 7, line 9. referenced correctly as Sherwin et al 2007. Elsewhere in the paper and in the reference list it is incorrectly noted as Sherwin et al 2008.

Response: Noted

Changes in the manuscript: The citation will be corrected to Sherwin et al. (2008).

(30) Page 7, Line 16. I don't find the more vigorous transport in the early part of the year is 'clearly evident' in this figure. A monthly plot might demonstrate this better.

Response: We agree that this is not easy to discern in Fig. 4, although it is evident in the mean seasonal cycle.

Changes in the manuscript: In the text (p.7, line 16), we will refer to a figure of monthly mean transport for each section, to be included in Supplementary Material.

(31) Page 7, Line 17. Is it possible to quote decadal mean figures rather than relying on our 'general impression'

Response: We agree that it is appropriate to quantify this statement.

Changes in the manuscript: We will cite in the text decadal-mean transports for 1988-97 and 1998-2007 (for each section), as suggested.

(32) Page 12, Line 13. Some statistics could be used to determine if this 'impression of generally smaller differences' is correct

Response: This statement is supported by the anomalies in annual-mean sea level difference shown in Fig. 12b, but we can provide 1988-97 and 1998-2007 averages of sea level difference to quantify the statement.

Changes in the manuscript: Decadal averages will be noted in the text.

(33) Page 13, Line 4. It would be good to know how much smaller - please quote some numbers to back up your statement.

Response: SSH differences during the 1990s typically vary in the seasonal range from 10 cm (summer) to 25 cm (winter); since 2000, summer differences fall as low as 5 cm while winter differences rarely exceed 20 cm; SSH differences are thus smaller by around 5 cm. We will quantify our statement more thoroughly on revision.

Changes in the manuscript: Accordingly, we will elaborate on the extent to which SSH differences have diminished since the 1990s. The extent to which these differences relate to changes in Slope Current transport will be revealed through our further attention to the utility of the SSH proxy (see response to specific comment above).

(34) Page 15, Line 9. Please put a date/timescale on 'recent'. It might not be recent when someone reads this next.

Response: The AMOC decline to which we refer is between the early 1990s and the mid 2000s (Balmaseda et al. 2007; Grist et al. 2009), as mentioned elsewhere in Sect. 4.

Changes in the manuscript: We will clarify the text accordingly.

(35) Page 15, Line 11. It might be better to say that this study provides "an estimate of the fraction" or "quantifies the fraction".

Response: Agreed

Changes in the manuscript: Revised as suggested

(36) Page 15, Line 13. starting "Existing EEL observations" These final two sentences feel to me oddly tagged on to the end and of little to do with the conclusion of this paper. If needed, they should be presented within the discussion.

Response: Agreed

Changes in the manuscript: In the revised discussion, we will review the need for these statements, and – if necessary – re-locate the text to discussion of transport across the EEL/OSNAP-East section.

References: McKenna, C., Berx, B. & Austin, W.E.N. (2016) The decomposition of the Faroe-Shetland Channel water masses using Parametric Optimum Multi-Parameter analysis. Deep-Sea Research Part I: Oceanographic Research Papers, 107: 9-21. http://dx.doi.org/10.1016/j.dsr.2015.10.013

---

## Author Response (AR1)

**Responses to Referees**

To each comment (normal font), we respond and specify changes in the manuscript (bold font), followed by a marked-up version of the revised manuscript (specified and other minor changes underlined). As a consequence of substantial changes in the main text, note that we have also revised the Abstract accordingly.

**Anonymous Referee #1**

General Comments:

The manuscript investigates the driving forces of the European Slope Current and also the related inflow of Atlantic water into the northwestern North Sea. In this study, a well-selected combination of observational data
10  and model results is employed. These data are integrated by means of innovative analytical methods, leading to new insights into the acting mechanisms. The overall impression is that the paper is carefully written in a clear and concise way. I just have some minor comments, which are given below. Therefore, altogether, I could recommend the manuscript for publication after a minor revision.

**Response: We thank the reviewer for supporting and thoughtful comments.**

15  **Changes in the manuscript: The manuscript has been revised in accordance with the comments below, alongside more extensive revisions in response to Referee 2.**

Detailed Comments:

Page 3, line 19: ". . . are usED"

**Response: Noted**

20  **Changes in the manuscript: Corrected**

Page 3, Section 2.2, model hindcast: It remains unclear, why ORCA12-N01 data are used at all. If ORACA12-N06 covers a longer period, I do not understand why you do not solely use this data set. Moreover, you must provide more support that a spatial resolution of approx. 10 km and a temporal resolution of 5 days are sufficient to describe the relevant processes related to the slope current variability.

25  **Response: The ORACA12-N06 hindcast was not available until after the long-established ORCA12-N01,**

and the offline trajectory calculations were until recently only possible with locally archived data (ORACA12-N06 data are archived remotely).

**Changes in the manuscript: In Sect. 2.2 (now Sect. 2.6), we discuss the extent to which spatial and temporal resolutions are appropriate to a study of the Slope Current dynamics and variability thereof: "With the barotropic Rossby radius at 55°N ranging from ~375 km (water depth 200 m) to ~1200 km (water depth 2000 m), the horizontal resolution of ORCA12 will comfortably resolve large instabilities and eddies associated with the Slope Current, although with corresponding baroclinic Rossby radii in the range 5-10 km, smaller-scale variability cannot be resolved. In the vertical dimension, there are 75 vertical levels, with 46 in the upper 1000 m, resolving the surface and bottom boundary layers that play an important role in Slope Current dynamics." (p.6, lines 5-9).**

Page 6, line 4: The argument that the number of drifters is limited does not really hold, since it would be possible to start the simulated particles exactly at the same time and place as in the drift experiment. Furthermore, the argument that sub-mesoscale processes hamper a proper comparison between observed and simulated tracers is in contradiction with the statement made in section 2.2, i.e., that eddy-resolving model data are employed.

**Response: In our statement, we mean that we routinely simulate 630 particle trajectories across a broad depth range representative of the Slope Current (per release), compared to 21 drifters that were drogued to drift with currents at 50 m in the Shelf Edge Study (SES) of LOIS. It would be possible to simulate specifically this number of particle trajectories, with start locations and times as in SES and drifting with currents at the same depth, and we could attempt this. However, the results would not add useful insight, not least because the ocean is chaotic at the mesoscale, so only in statistical terms (i.e., with a larger number of trajectories) are drifts in the ORCA12 hindcast representative of those in the real world. The purpose of Fig. 1 and Sect. 3.1 is to demonstrate that representation of the Slope Current in the ORCA12-N01 hindcast is broadly realistic in terms of pathways and timescales. Regarding sub-mesoscale processes and the point made in Sect. 2.2, we should clarify that ORCA12 resolves only mesoscale processes (10-100 km), but cannot resolve sub-mesoscale processes (1-10 km) such as inertial currents, frontal instabilities, etc.**

**Changes in the manuscript: In Sect. 3.1, we clarify the purpose of showing model trajectories alongside drifter tracks, and explain more clearly why we would not expect agreement: "Given the chaotic nature of mesoscale variability, we further note that pathways inferred from a more limited number of drifters are less statistically significant" (p.7, lines 27-28). We clarify the distinction between resolved mesoscale and unresolved sub-mesoscale processes: "… we show LOIS-SES drifter data alongside example model particle**

**trajectories, with the caveat that variability on length scales below ~10 km and time scales shorter than ~10 days are unresolved in the latter" (p.7, lines 15,17). We chose not to show individual model particle trajectories from the same start locations and times as SES.**

Page 8, line 16: Please clarify how $h_s$ and H are defined and give a reasoning why you distinguish between shelf and deep ocean. Actually, the Slope Current, which is in the focus of this study, is located at the transition between these two regions. Which equation holds in these transition areas?

**Response: Following Simpson and Sharples (2012), we introduce $h_s$ and H in Sect. 3.3.1, following the development of Equation (5) for the meridional sea surface slope in relation to water depth and the meridional density gradient. The reasoning for a distinction between the shelf and deep ocean is in the context of (5), which predicts the scenario presented in Fig. 16, with a steeper (downward to the north) slope in the deep ocean compared to the shelf, central to our understanding of the Slope Current. Given that the Slope Current is found at a water depth intermediate between $h_s$ and H, we would expects an intermediate meridional sea surface slope, but the key issue regards the growing difference between on and off shelf sea surface height with progression to the north, hence strengthening the associated geostrophic flow in the Slope Current.**

**Changes in the manuscript: We already provide this reasoning for $h_s$ and H in the manuscript.**

Page 8, line 25: Related to the previous comment, I do not see why H is always much larger than $h_s$.

**Response: H is always much larger than $h_s$ by definition, as H is the depth of the deep ocean while $h_s$ is the depth of the adjacent shelf sea.**

**Changes in the manuscript: No change necessary**

Page 12, line 4: The argument here is extremely questionable. If the number of particles entering the North Sea is well correlated with the Slope Current, there is no reason to assume that the definition of the Slope Current is not adequate, when it is correlated with salinity anomalies, since these should also be directly affected by the inflow of Atlantic water.

**Response: Referee 2 raises related points, and we expand on that response here. Contrary to the understanding so far established, we should not neglect variations in salinity, including contributions to Atlantic inflow excluded from our definition of the Slope Current. We address changing salinity in the model via the salinity along each trajectory (already calculated), and we will further seek observational**

**evidence for any changes.**

**Changes in the manuscript: We have extensively developed Sect. 3.4, substantiating statements that relate to the Slope Current influence on North Sea salinity. We find that downward salinity drifts in the model limit useful analysis of salinity as a tracer of Atlantic Water in the North Sea, although we recognize the contributory changes in both transport or salinity, distinguishing between "anomalous volume transport of mean salinity" and "mean volume transport of anomalous salinity". Indirectly, our analysis suggests that, during the 1990s, increasing salinity in the core of Atlantic Water was compensated by declining volume flux in the Atlantic inflow, consistent with little change in salinity of the northern North Sea (see p.18, lines 14-18).**

Page 13, line 3: Again, I question why ORCA12-N01 data are used at all, if the ORCA12-N06 data are more reliable as stated here, and cover even a longer period as mentioned earlier. To my opinion, this is unnecessary and just confuses the reader.

**Response: We do not claim that ORCA12-N06 data are more reliable, only that there is a good agreement in SSH variability between hindcasts over the period of overlap (see Fig. S9). As explained in an earlier response, the ORACA12-N06 hindcast was not available until after the long-established ORCA12-N01, and the offline trajectory calculations were until recently only possible with locally archived data (ORACA12-N06 data are archived remotely). While it may seem desirable to repeat the trajectory calculations with ORACA12-N06, these involve considerable effort and would severely delay manuscript revisions, while not substantively changing the conclusions of this study. We anticipate further studies of Slope Current variability that will involve the use of new and longer hindcasts. We chose to use ORCA12-N06 for the analysis in Sect. 3.5, as it is relatively straightforward to obtain model SSH data for direct comparison with the tide gauge data over the longer period.**

**Changes in the manuscript: At the end of Sect. 2.2 (new Sect. 2.6), we further clarify why we did not repeat trajectory analysis with the new ORCA12 hindcast: "While it would be instructive to also calculate particle drift and dispersal with the longer hindcast, such calculations are not straightforward with the remotely archived ORCA12-N06 datasets." (p.6, lines 21-22).**

Page 14, line 23: "Changes OF inflow ...."

**Response: Noted**

**Changes in the manuscript: Corrected**

**Anonymous Referee #2**

General Comments:

This is a nicely written and well set out paper and I found it easy to follow. The main results from the modelling study have the potential to add to our understanding of the drivers of North Sea inflow. However my main concern is in the analysis of North Sea inflow and the development of the SSH proxy, this aspect of the paper is poorly developed and I don't feel that the conclusions are well supported. I feel there was limited effort made to validate the model observations, and as a result, that sections 3.4 and 3.5 in particular need some revision.

**Response: We thank the reviewer for broad support and a very thorough review.**

**Changes in the manuscript: The manuscript has been extensively revised, paying particular attention to Sections 3.2, 3.4 and 3.5.**

(1) The use of the SSH metric is puzzling. The authors note that previous researchers have failed to establish a Shetland current transport series based on tide gauge records (Page 2, Line 8) but then fail to return to the subject as promised. This statement should be revised to be more accurate; but more preferable would be a more explicit discussion of this result in Section 4. More importantly there is no mention in this paper of the established pattern of Faroe-Shetland Channel circulation, particularly the recruitment of an additional branch of Atlantic water from the Western Atlantic through the Faroe Bank/Wyville Thomson Ridge region and the recirculation of the North Faroe Current in the channel. Recent estimate of transport through the channel using altimeter data (Berx et al., 2013) demonstrates for example, why a simple SSH metric might not be so useful.

**Response: We appreciate this comment, and agree that further discussion and clarification is necessary.**

**Changes in the manuscript: We have extended the Introduction (p.2, lines 10-32), the Results (Sect. 3.5), and the Discussion (p.18, line 28 to p.19, line 12) accordingly.**

(2) Logically we would expect some relationship between SSH variability and slope current but it would be worth examining some of the different mechanisms for transport of water into the NNS in the NE and the NW for example. This paper could be improved if there was some examination of the relationship between stronger slope current and/or stronger on-shelf transport and/or stronger North Sea inflow - the links between these three transports could be the key focus of the paper.

**Response: We agree that further analysis may provide helpful insight as to the relative contributions of Slope Current (as currently defined – see Sect. 3.2) and on-shelf transport to North Sea inflow.**

**Changes in the manuscript: We implicitly diagnose transport inshore of the Slope Current as a residual between sections from mainland Scotland to the Faroes, and between Shetland and the Faroes. We further diagnose North Sea inflow via the Fair Isle Current across the western JONSIS section. New results are presented in Figures 9, 10 and 14. We have accordingly revised Sect. 3.4 (p.13, lines 30 to p.14, line16).**

(3) There is no statement in the methods of how the particle backtracking was performed. The first mention of backtracking is in the results section. Also the depth distribution of the tracked particles within the North Sea is not described/discussed. This is of concern because the main Atlantic inflow in NW North Sea is known to flow at depth below the Baltic outflow.

**Response: ARIANE (as other tracking methods) can be used in backward mode, as forward mode, simply by time-reversing the analytical calculation of particle progress through grid-cells. Depth per trajectory is recorded. Depth distributions within the North Sea will be examined, to establish whether indeed inflow in the NW North Sea in ORCA12 is found below the fresh Baltic outflow.**

**Changes in the manuscript: This is explained in Sect. 2.3 (new Sect. 2.7): "We also use ARIANE in "backward" mode, which simply reverses (in time) the analytical calculation of particle progress through grid-cells, to examine the source of particles recruited to the Slope Current." (p.6, lines 31-32). Particle depth distributions are plotted as for particle density and age in revised Figures 2 and 3. Sect. 3.2 is revised accordingly (p.6, lines 25-31, lines 14-19).**

(4) Following on from this, it is not clear why the authors chose to examine surface (only) salinity data within the North Sea (only). Changes in salinity within the North Sea are not just linked to a change in transport of Atlantic Water but also the changing properties of the water that is transported. There was a strong trend of salinification in Atlantic water over the period of the 1990's. This trend is well documented and the reported/observed trend in the NNS is also one of salinification. Having modelled data should offer the authors a chance to investigate these trends and relationships. I believe that doing this would really add value to the paper. Alternatively can the authors explain why they chose surface salinity? Surely the surface salinity metrics are most likely to show the variability of freshwater flows.

**Response: Our choice to analyse only surface salinity within the North Sea is admittedly limiting, and we agree that a more analysis is warranted here. Indeed, salt transport anomalies may be partitioned between "anomalous volume transport of mean salinity" and "mean volume transport of anomalous salinity".**

**Changes in the manuscript: In keeping with our focus on the Slope Current, we now use the salinity along particle trajectories, with average particle salinity for forward trajectories shown added to Fig 2 (Fig.**

**2g,h). Considering the observational evidence for increasing salinity trends in the core of Atlantic Water at the Faroe-Shetland Channel alongside a reverse trend in the hindcast (Fig. 12c), we suspect that observed trends are associated with an additional influence from oceanic Atlantic Water that is not well represented in the hindcast (p.15, lines 5-7). However, we do consider that observed salinity trends may have broadly**

5   **compensated for a decline in the relatively saline Atlantic inflow (to the North Sea) during the 1990s, resulting in no major salinity trends within the northern North Sea during this period (p.18, lines 14-18). Sect. 3.4 is revised accordingly (p. 14, line 25 to p.15, line 11). Regarding the depth of Atlantic inflow to the North Sea, we further analyse the trajectory data and show that particles occupy a range of depths in the North Sea (see Fig. 2e,f).**

10   (5) The authors appear surprisingly unaware of the long time-series of data available, both in the northern North Sea and in the Faroe Shetland Channel where there are both hydrographic observations and long-term transport estimates. And also the Ellett Line. Before extracting salinity metrics from the North Sea it might be valuable to check how the model represents other observed patterns of salinity.

**Response: Thank you for pointing out the available observations. Our awareness of this data is implicit in**

15   **citation of Sherwin et al. (2008), but we will indeed look to model validation alongside observed transport estimates and salinity variations, as published. We obtained relevant data provided online with the ICES Report on Ocean Climate (http://ocean.ices.dk/iroc/).**

**Changes in the manuscript: Of the ICES data, we plot and use (in model evaluation) annual means of the salinity in the core of the Atlantic Water flowing through the Faroe-Shetland Channel (Fig. 12c), the mean**

20   **salinity in the Fair Isle Current (Fig. 12d), and monthly means of Atlantic inflow to the North Sea that are based on regional modelling (Fig. 10a).**

(6) From my understanding of these observations, I am not convinced that there is evidence of a decline in salinity during the 1990's in the North Atlantic-influenced regions of the northern North Sea. Curiously, the modelled decline in salinity almost mirrors the actual increase in salinity observed over the same period in the

25   Slope Current regions and the central North Sea.

**Response: On examination of ICES data, we agree salinity has not been observed to decline in the northern North Sea, although there is no clear evidence for an increase (while it has increased in much of the Slope Current).**

**Changes in the manuscript: Sect. 3.4 is revised accordingly - see response to previous Comment (4).**

(7) I would suggest the following observational evidence be examined.

B. Berx, B. Hansen, S. Østerhus, K. M. Larsen, T. Sherwin, and K. Jochumsen, 'Combining in-Situ Measurements and Altimetry to Estimate Volume, Heat and Salt Transport Variability through the Faroe Shetland Channel', Ocean Sci. Discuss., 10 (2013), 153-95.

5  Larsen, K. M. H., Gonzalez-Pola, C., Fratantoni, P., Beszczynska-Möller, A., and Hughes, S. L. (Eds). 2016. ICES Report on Ocean Climate 2015. ICES Cooperative Research Report No. 331. 79 pp

N. P. Holliday, S.L. Hughes, S. Bacon, A. Beszczynska-Möller, B. Hansen, A. Lavín, H. Loeng, K.A. Mork, S. Østerhus, T. Sherwin, and W. Walczowski, 'Reversal of the 1960s to 1990s Freshening Trend in the Northeast North Atlantic and Nordic Seas', Geophysical Research Letters, 35 (2008).

10  S. Dye, N.P. Holliday, S.L. Hughes, M. E. Inall, K. Kennington, T.J. Smyth, J. Tinker, O. Andres, and A. Beszczynska-Moller, 'Impacts of Climate Change on Salinity', in MCCIP Science Review, www.mccip.org.uk/arc, ed. by MCCIP, 2013).

N.P. Holliday, S.L. Hughes, S. Dye, M. E. Inall, J. Read, T. Shammon, T. Sherwin, and T.J. Smyth, 'Salinity in MCCIP Annual Report Card 2010-11', in MCCIP Science Review, 16pp. www.mccip.org.uk/arc, ed. by MCCIP,
15  2010).

Data from the Feie-Shetland, JONSIS and UTSIRE sections can be obtained from the host institutes or extracted from databases such as BODC, ICES and WODC.

**Response: Thank you for drawing these publications and data to our attention.**

**Changes in the manuscript: Berx et al. (2013) is cited on p.2 (lines 11, 16, 18, 21). Larsen et al. (2016) is**
20  **cited on p.17 (lines 13, 14). Holliday et al. (2008) is cited on p. 18 (line 11). As mentioned above, we use selected ICES observations and model estimates – see response to Comment (5).**

Specific comments:

(8) Page 2, Line 11. I'm not convinced that there is "some evidence of surface freshening through the 1990's". Established time series either show no trend in salinity or a slight increase during this period, following the
25  increases seen in Atlantic Water. The freshening that has been observed is limited to the southern North Sea.

**Response: This statement was based on inspection of figures in Højllo et al. (2009), as indicated. We accept**

the referee's point about different salinity changes elsewhere, and we have re-visited the broader patterns and character of salinity change.

**Changes in the manuscript: In the Introduction, we now restrict our remarks to the observed warming of the North Sea (p.2, line 33 to p.3, line 4).**

(9) Page 7, Line 6-8: The definition of the sections is based on the slope current in the model, but what criteria was applied to define the slope current? See also comments relating to the area of Atlantic water.

**Response: The Slope Current was identified as a narrow band of high velocity current confined between isobaths, in 5-day averaged ORCA12 datasets.**

**Changes in the manuscript: In Sect. 3.2, we clarify that the Slope Current is "identified as a narrow band of high velocity ($>10$ cm s$^{-1}$) in 5-day mean fields" (p.6, lines 28,29), for the purposes of calculating transport.**

(10) Page 7, Line 21 onwards: The authors fail to acknowledge the established knowledge of Faroe-Shetland Channel circulation, particularly the recruitment of an additional branch of Atlantic water from the Western Atlantic through the Faroe Bank/Wyville Thomson Ridge region and the re-circulation of the North Faroe Current in the channel. These offer some explanation as to why a single index of SSH might not represent slope current variability.

**Response: Thank you for raising this point, which we should certainly bear in mind, in regard to the drivers of variability in a SSH slope index.**

**Changes in the manuscript: We recognize this issue in the Discussion (p.19, lines 5-7).**

(11) Page 11, Line 27. The strong freshening trend from your model data is in the surface salinity of the NW North Sea. Terschelling is a station in the central North Sea and as presented, offers little in support of your observations. The Utsire station as presented by Hjollo sits well within the Baltic Outflow which would link much more strongly to your NE time-series and as mentioned before is likely to mostly reflect variability of freshwater input.

**Response: Thank you for pointing this out. As explained in response to earlier comments, we have revised and reduced our analysis and interpretation of salinity anomalies.**

**Changes in the manuscript: Sect. 3.4 has been revised accordingly.**

(12) Page 11, Line 34 to Page 12, Line 4: Could the authors elaborate on the spatial variability of what would be their defined Atlantic water mass to highlight what this impact may be?

**Response: In the Faroe-Shetland Channel (FSC), salinity maxima are evident in particle-mean salinity, in the range 35.45-35.60 psu. Water of this salinity corresponds to the "North Atlantic Water" of salinity around 35.42 psu that dominates the upper 200 m on the Shetland side of the FSC, recently identified by McKenna et al. (2016). Moving onto the shelf, particle-mean salinity declines to 35.20-35.35 psu. This is consistent with mixing of Atlantic Water and relatively fresher water on the shelf, for which salinity ranges 35.00-35.25 psu at the Ellett line (e.g. Fig. 14 in Inall et al. 2009).**

**Changes in the manuscript: Referring to mean salinity in Fig. 2g,h, we describe the variation of salinity from the Slope Current onto the shelf in the Faroe-Shetland Channel (p.8, line 32 to p.9, line 5).**

(13) Page 12, I'm worried about these correlations. All three time-series have a similar downward trend and hence they would always be reasonably correlated. Has this been accounted for in the correlation analysis? If not then by the same argument the wind forcing metric that you include in Figure 8 would also be correlated.

**Response: Trends are included in the correlated data. We agree that there is limited value in correlations between short time series (given only 20 annual values) that are characterized by similar trends.**

**Changes in the manuscript: We omit correlations from Sect. 3.4. We retain the correlations in Sect. 3.5 that relate to high frequency time series (between sea level differences and transports in ORCA12-N01), and between sea level differences over the longer period (based on ORCA12-N06 and tide gauges).**

(14) Page 12, Line 4. Have you examined how the salinity of the slope water has increased over the period? Is this reflected in your model? During the periods of weaker modelled transport we know that the slope current became saltier.

**Response: We address changing salinity in the model via the salinity along each trajectory (already calculated), and we have sought observational evidence to evaluate the changes. In summary, we find some support in observations (GODAS analyses) for a freshening trend in the Slope Current at FASTNEt and EEL sections from the early 1990s to the mid 2000s, but less agreement at the Shetland Slope section, where the freshening trend in ORCA12 is opposite to an increasing salinity trend in the observations. We suspect that much of the observed trend is attributed to more oceanic origin (rather than anomalies advecting with the Slope Current from further south). We further find some agreement at the JONSIS line for a slight freshening trend in both observations and hindcast, although local processes (missing from the**

**model) likely dominate observed interannual variability here. Regarding the link between transport and salinity, Slope Current salinity declines with transport in the model, but observations are not sufficient to verify this relationship – indeed, no apparent increase in ICES-modelled Atlantic inflow coincide with remarkable recent increases in JONSIS salinity (in GODAS analyses).**

5 **Changes in the manuscript: We use ICES reported salinity data for the Shetland Slope and JONSIS sections, along with data from the GODAS ocean analysis product at all four sections (Figs. 12, S11, S12). We have developed Sect. 3.4 accordingly (p.14, line 25 to p.15, line 19).**

(15) Page 12, Lines 15-17: Are the seasonal cycles in the Shetland Slope transport and SSH differences in phase?

**Response: Shetland Slope transport and Lerwick-Torshavn SSH differences are clearly in phase,**
10 **comparing Fig. 4c and Fig. 13b, as are Wick-Lerwick SSH differences and Fair Isle Current transports.**

**Changes in the manuscript: We emphasize this in-phase relationship in Sect. 3.5 (p. 15, lines 21-23).**

(16) Page 13, Line 9. It would be nice to see some comparison of the modelled slope current with some of the observations of the slope current and more sophisticated estimates from altimeter data.

**Response: We are aware of transport estimates for the Atlantic inflow between the Faroes and Shetland**
15 **since December 1992 (Fig. 10 in Berx et al., 2013) in the range 1-5 Sv. Extending our "Shetland Shelf" section, we find considerably higher transport of 7.04 ± 2.18 Sv.**

**Changes in the manuscript: The revised transport estimates are plotted in Fig. 4c. Where we introduce Slope Current transport data for the Shetland Shelf, we note that these transports "… at the upper end of observed ranges (e.g., Sherwin et al., 2008), and considerably higher than "Atlantic inflow" estimates of 2.7**
20 **± 0.5 Sv in the Faroe-Shetland Channel (Berx et al., 2013)" – see p.9, lines 27,28.**

(17) Page 14, Line 11. ref to limited observations. The Northern North Sea is relatively rich in observations, being sampled across two sections (Feie-Shetland and JON- SIS/Utsire) least 5 times per year by UK (Scotland) and Norway. These data are publically available for you to use in your research.

**Response: We have accessed the necessary data, as recommended.**

25 **Changes in the manuscript: We outline selected data in new Sect. 2.1: "The following data are provided as part of the ICES Report on Ocean Climate (IROC), available at http://ocean.ices.dk/iroc:**

- **Depth-averaged inflow and outflow to/from the North Sea, centred on 59°N, 1°E, as modelled volume transport between Orkney (Scotland) and Utsira (Norway), monthly averaged from January 1985**
- **Salinity in the Fair Isle Current, centred on 59°N, 2°W, averaged over the depth range 0-100 m, irregularly-sampled and annually-averaged from 1960**
- **Salinity for the Faroe Shetland Channel – Shetland Shelf, centred on 61°N, 3°W, the maximum in the upper layer high salinity core, sampled 3 times per year (April/May, September/October and December) from 1950**

**We further sample monthly-mean salinity in the NCEP Global Ocean Data Assimilation System (GODAS) analysis fields spanning 1980-2016 (NOAA Climate Prediction Center, see http://www.cpc.ncep.noaa.gov/products/GODAS/). These time series data are used to evaluate time series of similar quantities in the ORCA12-N01 hindcast and derived Lagrangian data." (p.4, lines 5-16). We plot these data alongside hindcast data in Figs. 10 and 12.**

(18) Page 15, Line 6. Sentence starting "These transient events..." In which case presumably we would see a return to higher salinities - has anyone noticed this? I'm not sure you can evidence this statement.

**Response: This is a good point. We have examined the GODAS ocean analysis for evidence of increasing salinity in the Slope Current. At the FASTNEt and EEL sections, salinity at around 800 m increased by around 0.05 psu over much of 2014-16 (Fig. S11a,b). Regarding the possibility of concurrent Slope Current strengthening, we have used the thermal wind relation to predict an approximate doubling of eastward geostrophic transport at 30°W in mid-latitudes, associated with increased meridional density gradients due to subpolar cooling. This work is in progress, for a possible future manuscript.**

**Changes in the manuscript: We have added text to the Discussion in accordance with this evidence and these preliminary findings (p.19, lines 21-25).**

(19) Page 15, Line 13. Reference is also made here to observed trends in the Ellett line data, this raises the question of why the authors did not examine/compare the trends in salinity from the EEL observations as these also show an increase in salinity during the 1990's?

**Response: The GODAS data provide observations at our EEL section. Repeating an earlier response, we address changing salinity in the model via the salinity along each trajectory (already calculated), which we can compare to the GODAS data. The freshening trend in ORCA12 is also seen, although to a lesser extent, in the GODAS data, with peak salinities in the early 1990s, followed by some decline towards to**

**mid 2000s. It may be that salinity trends in the Slope Current are distinct from increasing salinity trends seen across the wider EEL.**

**Changes in the manuscript: Comparisons between the hindcast and observations are provided in new Fig. 12, with the EEL comparison in Fig. 12b. Additional Figs. S11 and S12 illustrate GODAS salinity time series across a range of depths. We discuss observed and simulated salinity changes, over 1988-2007, in the Slope Current to the west of Scotland in Sect. 3.4 (p. 14, line 25 to p.15, line 11).**

(20) In methods/supplementary material:

You examine average surface salinity values over relatively large box areas (NW and NE). The size of these regions makes it likely that both of salinity time-series are capturing variability in Norwegian Coastal current. This is evidenced by the salinity minima in summer in this region. Away from the regions of Baltic influence, in the northern North Sea we would expect salinity maxima in Autumn (September) as a result of Atlantic Inflow. I think it might be valuable to consider different regions and examine data at depth - not just surface values?

**Response: Thank you for this insight. It would be appropriate to re-calculate salinity at various depths, for re-defined, perhaps irregular, sub-regions that are informed by the pattern of inflow evident in Fig. 2. However, as we have shifted our focus on salinity towards the Slope Current and Atlantic inflow (Sect. 3.4), it was decided to remove the analysis of North Sea salinity in ORCA12, taking into consideration that the model does not represent some processes in shelf seas that strongly influence salinity distributions (tidal mixing, interannual variability of runoff).**

**Changes in the manuscript: We have replaced Fig. 10 with time series of salinity anomalies in the upstream Slope Current and Atlantic inflow (see previous responses). The text in Sect. 3.4 is revised accordingly. We have removed Fig. S8 (area-averaged surface salinities in the North Sea).**

Minor/Technical comments:

(21) The most common spelling is Faroe and not Faeroe.

**Response: Accepted**

**Changes in the manuscript: Text is revised accordingly**

(22) Fig 1, Lerwick is not accurately positioned in Figure 1d. The position of the Shetland Slope transect (3) seems to have moved between 1c and 1d.

**Response: Accepted**

**Changes in the manuscript: Fig. 1c,d has been revised accordingly**

(23) Fig 1, I find scales hard to read on Figures 1a and 1b, the font size used in 1c and 1d is more reasonable.

**Response: Accepted**

**Changes in the manuscript: Fig. 1 is enlarged in the revised manuscript.**

(24) Fig 2, I note a reference to the log scale - but I think the scale used could be described more clearly.

**Response: Agreed**

**Changes in the manuscript: We elaborate in the caption: "The logarithmic scale for density, ranging from -4 to -1, equates to 0.01-10% of all particle positions".**

(25) Fig 5, the number of colours in the scale could align better to the scale intervals.

**Response: Point taken**

**Changes in the manuscript: We will re-plot the figure accordingly in a final version, if accepted for publication.**

(26) Page 3, Line 18: ...are used...

**Response: Noted (as Referee 1)**

**Changes in the manuscript: Corrected**

(27) Page 6, Line 1. the position of the FASTNEt, EEL and Shetland Shelf sections should be be defined. In section 2.3, floats are mentioned which were deployed for FASTNEt (but which aren't analysed in this manuscript and their position is not described). Is the "FASTNEt section" the line where the ORCA particles were released? If so, it needs to be stated on Page 4, Lines 8-10 that this is henceforth referred to as the FASTNEt section. In Table 1 the central locations of EEL and Shetland shelf are mentioned, but the length/endpoints of each section is not defined.

**Response: We originally calculated transport across the FASTNEt section where we release particles. We**

still calculate transport at this latitude, but across a somewhat wider section. We agree that full details of all transport sections should be defined.

**Changes in the manuscript: We now define the "FASTNEt release section" from which particles are tracked forwards and backwards (Sect. 2.7, p.6, lines 26-28). The endpoints for longer FASTNEt, EEL and Shetland Shelf sections are defined in Sect. 3.2 (p.9, lines 21-24) and the Table 1 caption, and all sections are now accurately indicated in Fig 1c,d.**

(28) Page 6, line 3. 'northwest' and 'northeast' sectors not adequately defined until 3.4 (Page 11, Line 15) please move that to here or put into methods.

**Response: This is an oversight.**

**Changes in the manuscript: We now define northwest and northeast sectors of the North Sea in new Sect. 2.7 (p.7, lines 6-7), where we introduce the Lagrangian methodology.**

(29) Page 7, line 9. referenced correctly as Sherwin et al 2007. Elsewhere in the paper and in the reference list it is incorrectly noted as Sherwin et al 2008.

**Response: Noted**

**Changes in the manuscript: The citation is corrected as Sherwin et al. (2008)**

(30) Page 7, Line 16. I don't find the more vigorous transport in the early part of the year is 'clearly evident' in this figure. A monthly plot might demonstrate this better.

**Response: We agree that this is not clear in Fig. 4, and we now revisit this statement.**

**Changes in the manuscript: In Sect. 3.2, we have moderated our statement as follows: "This is indicative of a somewhat more vigorous circulation during January-June, although monthly-mean transports at the three Slope Current sections and the JONSIS section (Fig. S6) do not provide conclusive evidence for this." (p.10, lines 2-4), referring to a new figure showing monthly-mean transports, in Supplementary Material.**

(31) Page 7, Line 17. Is it possible to quote decadal mean figures rather than relying on our 'general impression'

**Response: We agree that it is appropriate to quantify this statement.**

**Changes in the manuscript: We provide decadal-mean transports for 1988-97 and 1998-2007 (for each**

**section) in Table 1, and refer to these in Sect. 3.2 (p.10, lines 5-6).**

(32) Page 12, Line 13. Some statistics could be used to determine if this 'impression of generally smaller differences' is correct

**Response: We could provide 1988-97 and 1998-2007 averages of sea level difference to quantify the statement.**

**Changes in the manuscript: Following major revision of Sect. 3.5, we no longer discuss small reductions in sea level difference at this point, although we do discuss recent declines in the longer records – see response to next comment.**

(33) Page 13, Line 4. It would be good to know how much smaller - please quote some numbers to back up your statement.

**Response: SSH differences during the 1990s typically vary in the seasonal range from 10 cm (summer) to 25 cm (winter); since 2000, summer differences fall as low as 5 cm while winter differences rarely exceed 20 cm. SSH differences are smaller by around 5 cm in selected years. Averaging Lerwick-Torshavn sea level differences over the decade 1989-1998, and over 1999-2012, we find that differences are reduced by 2 cm in the more recent period, equivalent to a small transport reduction of around 0.5 Sv.**

**Changes in the manuscript: Average Lerwick-Torshavn sea level differences for 1989-1998 and 1999-2012, and the change, are now stated in Sect. 3.5 (p.16, lines 19,20).**

(34) Page 15, Line 9. Please put a date/timescale on 'recent'. It might not be recent when someone reads this next.

**Response: The AMOC decline to which we refer is between the early 1990s and the mid 2000s (Balmaseda et al. 2007; Grist et al. 2009), as mentioned elsewhere in Sect. 4.**

**Changes in the manuscript: In the revised Discussion, we no longer refer to recent weakening of the AMOC, but still remark on decline of AMOC in mid-latitudes between the early 1990s and the mid 2000s (p.19, lines 14-16).**

(35) Page 15, Line 11. It might be better to say that this study provides "an estimate of the fraction" or "quantifies the fraction".

**Response: Noted**

**Changes in the manuscript: We no longer make this statement in the revised manuscript.**

(36) Page 15, Line 13. starting "Existing EEL observations": These final two sentences feel to me oddly tagged on to the end and of little to do with the conclusion of this paper. If needed, they should be presented within the discussion.

**Response: Agreed**

**Changes in the manuscript: These last two sentences are re-located earlier in the discussion, to p.18 (lines 23-27) and p.17 (lines 17-19).**

**References:**

[revised manuscript text omitted]